# Unsupervised discovery of the shared and private geometry in multi-view data

## Abstract

Studying complex real-world phenomena often involves data from multiple views (e.g. sensor modalities or brain regions), each capturing different aspects of the underlying system. Within neuroscience, there is growing interest in large-scale simultaneous recordings across multiple brain regions. Understanding the relationship between views (e.g., the neural activity in each region recorded) can reveal fundamental insights into each view and the system as a whole. However, existing methods to characterize such relationships lack the expressivity required to capture nonlinear relationships, describe only shared sources of variance, or discard geometric information that is crucial to drawing insights from data. Here, we present SPLICE: a neural network-based method that infers disentangled, interpretable representations of private and shared latent variables from paired samples of high-dimensional views. Compared to competing methods, we demonstrate that SPLICE **1)** disentangles shared and private representations more effectively, **2)** yields more interpretable representations by preserving geometry, and **3)** is more robust to incorrect a priori estimates of latent dimensionality. We propose our approach as a general-purpose method for finding succinct and interpretable descriptions of paired data sets in terms of disentangled shared and private latent variables.

## 1 Introduction

Given multiple high-dimensional datasets that each capture a different view of a single underlying system, gaining deep insight into the system requires understanding the information that is common (shared) and unique (private) between views. Examples include identifying the shared semantic overlap between text captions and images (Lee & Pavlovic, 2021), integrating information from single-cell transcriptomics and proteomics to characterize cell state (Argelaguet et al., 2021), and performing sensor fusion in robots (Fadadu et al., 2022). This multi-view paradigm is becoming prevalent in neuroscience due to new recording technologies that are rapidly increasing the scale of neuronal recordings (Jun et al., 2017; Steinmetz et al., 2021). Cutting-edge neural recordings capture the simultaneous activity of many neurons across multiple regions, at single-neuron resolution. If we consider each brain region as a view into the underlying brain-wide activity, understanding what information is represented separately in individual regions or shared between them is vital to characterizing each region's functional role.

Since representations of local information (private to a region) and global information (shared across regions) can be nonlinearly multiplexed with each other, an understanding of each requires *disentangling* the two information types. Mathematically, we can frame this task as an inverse problem where, for any sample of paired data points $x_A \in A$ and $x_B \in B$, there exist a shared set of latent variables $s$ and two private sets of latent variables $z_A$ and $z_B$. The high-dimensional observations are then generated as $x_A = g_A(s, z_A)$ and $x_B = g_B(s, z_B)$ for two distinct nonlinear functions $g_A(\cdot)$ and $g_B(\cdot)$, where $z_A, s$ and $z_B$ are all statistically independent (Fig. 1a). Our goal is to find $\{g_A(\cdot); g_B(\cdot); z_A; s; z_B\}$.

Although many multi-view learning methods have been developed in the general machine learning literature, these methods primarily seek to create multimodal generative models, where individual factors of variation can be independently manipulated to produce realistic-looking data (Palumbo et al., 2023; Lee & Pavlovic, 2021). These methods typically impose an isotropic Gaussian prior on latent variables and use total-correlation objectives to encourage factorization and enable the desired

sampling of the latent space. These architectural choices work towards the models' stated goals of generation and disentangling, but destroy latent geometric structure vital to drawing insight from the data. Achieving *understanding* of the overall system, the primary goal in scientific machine learning, instead requires interpreting the latent representations themselves – their content, structure, and how they influence the observed data.

In neuroscience, for example, examining the latent geometry of neural representations has provided insight into the computations that single brain regions perform; manifold learning methods have revealed a ring geometry in the population activity of head direction cells, enabling blind discovery and decoding of the represented variable (Chaudhuri et al., 2019). Similar methods, applied to the neural activity of entorhinal grid cells, discovered a toroidal geometry that confirmed predictions from theoretical continuous attractor models (Gardner et al., 2022). Finally, geometry-preserving retinotopic maps have helped to delineate the borders of adjacent visual regions (Engel et al., 1994; Zhuang et al., 2017). Manifold learning methods (Tenenbaum et al., 2000; Silva & Tenenbaum, 2002; Roweis & Saul, 2000) typically use local distances in high-D observation space to estimate geodesic distances and then learn a low-dimensional embedding that preserves the estimated distances. However, applying manifold learning to multi-view data introduces complications; when shared and private information are mixed, the distances used by these methods reflect a combination of both types of info, preventing accurate estimation of the geometry of only the shared or private components.

Multi-view learning in neuroscience has typically relied on simpler, classical linear methods such as Canonical Correlational Analysis (Hotelling, 1936) and Reduced Rank Regression (Izenman, 1975) to identify shared latent variables between brain regions (Semedo et al., 2019; MacDowell et al., 2025; Ebrahimi et al., 2022) or align latent spaces across days (Gallego et al., 2020). However, linear models lack the expressivity necessary to model the complex nonlinear relationships between latent variables and neural activity. More recent nonlinear models aim to better capture these relationships, but lack explicit loss terms to disentangle shared and private latents, instead relying on the model architecture to implicitly encourage disentangling (Gondur et al., 2023; Sani et al., 2024). Such implicit disentangling is often too weak to produce statistically independent representations. Especially when the dimensionality of the shared space is mis-specified, such models can leak view-specific variance into the shared latents without penalty, or vice-versa. This vulnerability, also prevalent in the machine learning literature, is especially problematic for blind scientific discovery, where the true dimensionality of latent structure is unknown and mis-specification is unavoidable. (We discuss additional related work in neuroscience and machine learning in the Discussion.)

There thus remains a need for a method that can disentangle nonlinearly mixed shared and private latent variables without a priori knowledge of latent dimensionality, while retaining geometric structure that promotes interpretability. The primary contributions of our work are: 1) a new network architecture (SPLICE) that separates and captures both the shared and private intrinsic geometry of a multi-view dataset, 2) validation of the architecture in controlled simulations showing that our model achieves superior disentangling, interpretability, and robustness to mis-specified latent dimensionality than state-of-the-art methods (Lyu et al., 2021; Lee & Pavlovic, 2021), and 3) a real neural data example showing that our model blindly discovers known shared information, validating its utility for scientific discovery without the a priori hypotheses required by previous targeted studies.

## 2 SUBMANIFOLD PARTITIONING VIA LEAST-VARIANCE INFORMED CHANNEL ESTIMATION (SPLICE)

Given paired observations that represent two views of a single underlying system, SPLICE aims to infer disentangled latent representations of shared and private information that **also** preserve intrinsic submanifold geometry. Under the forward model described in Section 1, each set of latent variables, shared or private, corresponds to a submanifold of the overall data manifold in observation space (Fig. 1a, right). As one set of latent variables varies and the other is held constant, it traces out the corresponding submanifold in the high-D observation space. However, recovering the submanifolds directly from the original observations is difficult; the latent submanifolds are nonlinearly mixed in observation space, so local distances in observation space are influenced by both shared and private information. Therefore, we cannot use conventional manifold learning techniques that rely on these local distances. Instead, we must first isolate the submanifolds before applying conventional manifold learning techniques.

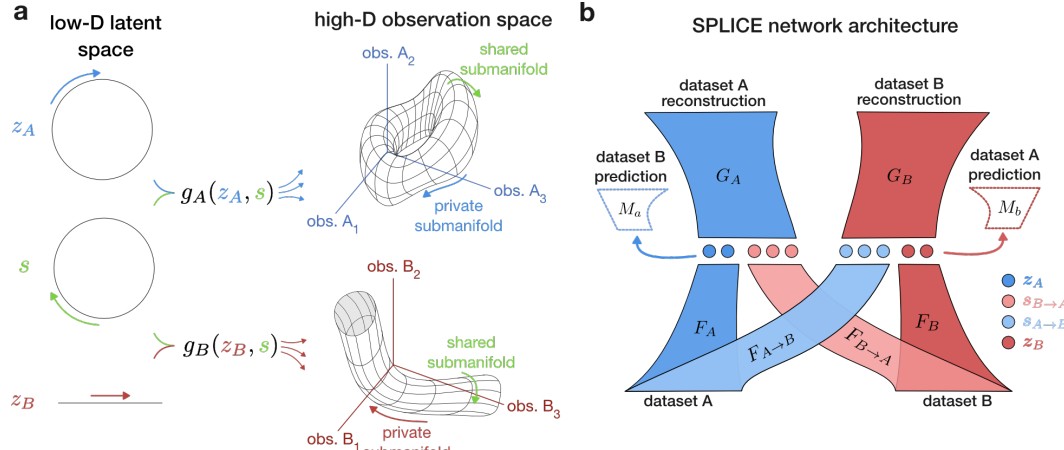

Figure 1: Problem formulation and model architecture. **a)** Illustration of the observation model. Low-dimensional private and shared latent variables are combined nonlinearly to form low-dimensional manifolds embedded in the $A$ view and $B$ view high-dimensional observation spaces. **b)** The SPLICE unsupervised autoencoder network architecture.

SPLICE adopts a two-step approach to address this requirement. In Step 1 we use predictability minimization (Schmidhuber, 1992) in a crossed autoencoder framework to learn disentangled representations of the shared and private latent variables. Assuming disentangling was successful, we can then hold one set of latents constant while varying the other to project data onto the shared or private submanifolds, and use conventional manifold learning techniques to estimate geodesic distances along them. Step 2 then finetunes the latent representations to preserve the estimated submanifold geometries, providing insight into the structure of the latent variables.

**Step 1: Disentangling private and shared latent variables:** SPLICE uses a symmetric autoencoder framework to infer latent variables. Each view is input into two encoders, one that generates a shared set of latents and the other a private set (Fig. 1b):

$$\underbrace{\widehat{\boldsymbol{z}}_A = F_A(\boldsymbol{x}_A)}_{\text{private A latent}}, \quad \underbrace{\widehat{\boldsymbol{z}}_B = F_B(\boldsymbol{x}_B)}_{\text{private B latent}}, \quad \underbrace{\widehat{\boldsymbol{s}}_{B \to A} = F_{B \to A}(\boldsymbol{x}_B)}_{\text{shared from B latent}}, \quad \underbrace{\widehat{\boldsymbol{s}}_{A \to B} = F_{A \to B}(\boldsymbol{x}_A)}_{\text{shared from A latent}} \quad (1)$$

Each view is reconstructed from its own private latents and the shared latents from the other view:

$$\widehat{\boldsymbol{x}}_A = G_A(\widehat{\boldsymbol{s}}_{B \to A}, \widehat{\boldsymbol{z}}_A), \qquad \widehat{\boldsymbol{x}}_B = G_B(\widehat{\boldsymbol{s}}_{A \to B}, \widehat{\boldsymbol{z}}_B). \quad (2)$$

The encoders $F_A$, $F_B$, $F_{A \to B}$, $F_{B \to A}$ and decoders $G_A$, $G_B$ are all parameterized as multi-layer neural networks. Using shared latents from one view to reconstruct the other view guarantees that a view's private information does not leak into the shared latents used to reconstruct it (Karakasis & Sidiropoulos, 2023). To prevent the other type of leakage – shared information into private latents – we turn to predictability minimization (Schmidhuber, 1992).

The intuition behind predictability minimization is that if a variable can predict something about another, there must be non-zero mutual information between them. We therefore introduce auxiliary "measurement networks" that try to predict each dataset as well as possible based on the other's private latent: $\boldsymbol{x}_A^{pred} = M_{B \to A}(\widehat{\boldsymbol{z}}_B)$, $\boldsymbol{x}_B^{pred} = M_{A \to B}(\widehat{\boldsymbol{z}}_A)$. We use these measurement networks in an adversarial disentangling scheme: in predicting the opposite region's observations as well as possible, the measurement networks try to exploit any shared information that has leaked into the private latents. If there is no shared information in the private latents, the best prediction the measurement networks can make (in an MSE sense) is to always output the mean of the target. Thus for well-trained measurement networks, we have

$$I(\widehat{\boldsymbol{z}}_B; \boldsymbol{x}_A) = 0 \to \text{Var}_{\boldsymbol{x}_B}[M_{B \to A}(\widehat{\boldsymbol{z}}_B)] = 0 \quad I(\widehat{\boldsymbol{z}}_A; \boldsymbol{x}_B) = 0 \to \text{Var}_{\boldsymbol{x}_A}[M_{A \to B}(\widehat{\boldsymbol{z}}_A)] = 0. \quad (3)$$

We thus train the private encoders to minimize $\text{Var}_{\boldsymbol{x}_B}[M_{B \to A}(\widehat{\boldsymbol{z}}_B)]$ and $\text{Var}_{\boldsymbol{x}_A}[M_{A \to B}(\widehat{\boldsymbol{z}}_A)]$ (in addition to reconstruction error) to encourage the measurement network predictions to be as poor as

possible. Predicting data observations rather than predicting inferred shared latents prevents shared information from leaking into the private latents, *regardless of whether that shared information is present in the inferred shared latents*. As we will show below, this makes our model more robust to mis-specified private latent dimensionality than existing methods.

**Step 1 loss functions and fitting:** We train the encoders and decoders $\theta_{ae} = \{G_A, G_B, F_A, F_B, F_{A\to B}, F_{B\to A}\}$ to minimize the reconstruction losses $\mathcal{L}_{rec}^A$ and $\mathcal{L}_{rec}^B$ and the variance in the measurement networks' outputs:

$$\mathcal{L}_{SPLICE} = \mathbb{E}[\overbrace{\|\boldsymbol{x}_A - \widehat{\boldsymbol{x}}_A\|_2^2}^{\mathcal{L}_{rec}^A} + \overbrace{\|\boldsymbol{x}_B - \widehat{\boldsymbol{x}}_B\|_2^2}^{\mathcal{L}_{rec}^B} + \lambda_{dis}\left(\text{Var}\left[M_{A\to B}(\widehat{\boldsymbol{z}}_A)\right] + \text{Var}\left[M_{B\to A}(\widehat{\boldsymbol{z}}_B)\right]\right)]$$
$$\theta_{ae}^* = \arg\min_{\theta_{ae}} \mathcal{L}_{SPLICE}. \tag{4}$$

Successful disentangling with predictability minimization requires well-trained predictors $\theta_{pred} = \{M_{A\to B}, M_{B\to A}\}$. We continuously update $\theta_{pred}$ to minimize the prediction losses as

$$\theta_{pred}^* = \arg\min_{\theta_{pred}} \mathbb{E}[\overbrace{\|\boldsymbol{x}_A - M_{B\to A}(\widehat{\boldsymbol{z}}_B)\|_2^2}^{\mathcal{L}_{pred}^A} + \overbrace{\|\boldsymbol{x}_B - M_{A\to B}(\widehat{\boldsymbol{z}}_A)\|_2^2}^{\mathcal{L}_{pred}^B}]. \tag{5}$$

To fit the multiple interacting networks comprising the SPLICE model, we adopt an alternating optimization approach (Schmidhuber, 1992) (Algorithm 1). Because our disentangling strategy relies on the measurement networks being well-trained and able to learn complex relationships, we use measurement networks that are as wide and deep as the decoder networks, and take multiple gradient steps to minimize the measurement prediction losses $\mathcal{L}_{pred}^A$ and $\mathcal{L}_{pred}^B$ for each single step of the other losses. [1]

**Step 2: Geometry Identification and Preservation:** With the disentangled shared and private latent representations from Step 1, Step 2 of SPLICE refines these representations to preserve the intrinsic submanifold geometries, which is crucial for interpretability. This process involves three sub-steps:

**Projecting onto Submanifolds:** To estimate the submanifold geometry of each latent space, we first use the trained network from Step 1 to project data points onto the respective submanifolds. E.g., to project points onto the private submanifold of view $A$ (associated with $\widehat{\boldsymbol{z}}_A$), we first select a random observation sample $\boldsymbol{x}_B'$, obtain its shared representation $\widehat{\boldsymbol{s}}_{B\to A}^{\text{fix}} = F_{B\to A}(\boldsymbol{x}_B')$. We then project the training data points onto the private submanifold by passing their $\boldsymbol{x}_A$ samples through $F_A(\cdot)$ and decoding with the fixed shared component from the sample $\boldsymbol{x}_B'$:

$$\widehat{\boldsymbol{x}}_A^{\boldsymbol{z}_A \text{ subm}} = G_A(\widehat{\boldsymbol{s}}_{B\to A}^{\text{fix}}, F_A(\boldsymbol{x}_A)) \tag{6}$$

Assuming the latent spaces are well-disentangled, this process will isolate the view A private submanifold in the observation space, since $\widehat{\boldsymbol{s}}_{B\to A}$ is held constant while $\widehat{\boldsymbol{z}}_A$ varies. We use a similar procedure to project points onto the shared submanifolds and the view B private submanifold.

**Estimating submanifold geodesic distances:** Given the data points projected onto the shared and private submanifolds, we can use traditional manifold learning techniques to estimate their geometry: for each set of submanifold projections, we construct a nearest-neighbors graph and estimate the geodesic distances (denoted $D_{A_{priv}}^{geo}$ for the view A private submanifold projections) from each data point to a small number of landmark points. Using landmarks rather than computing all pairwise distances significantly reduces the runtime complexity of this step, from $\mathcal{O}(N^2 log N)$ to $\mathcal{O}(nN log N)$, where $N$ is the number of points and $n \ll N$ is the number of landmarks (Silva & Tenenbaum, 2002). Thus, this step can take multiple orders of magnitude less time than the neural network training. As in traditional manifold learning, we can then constrain Euclidean distances in each low-dimensional embedding space to match the estimated geodesic distances from the corresponding observation-space submanifold projections.

**Fine-tuning with geometry-preserving loss:** The SPLICE autoencoders are fine-tuned by augmenting the original SPLICE loss function, $\mathcal{L}_{SPLICE}$ (Eq. 4), with terms that penalize discrepancies

---

[1]It has been noted (Goodfellow et al., 2014) that training of the measurement networks is different, but closely related, to Generative Adversarial Networks (GANs): the $M(\cdot)$ network aims to improve its prediction of a data set, while the $F(\cdot)$ network that provides $M$'s input aims to hinder this prediction.

between the estimated submanifold geodesic distances and the corresponding latent space Euclidean distances:

$$\theta_{ae}^* = \arg\min_{\theta_{ae}} \left[ \mathcal{L}_{\text{SPLICE}} + \lambda_{\text{geo}} \left( \mathcal{L}_A^{\text{geo}} + \mathcal{L}_B^{\text{geo}} + \mathcal{L}_{S_{B\to A}}^{\text{geo}} + \mathcal{L}_{S_{A\to B}}^{\text{geo}} \right) \right]. \tag{7}$$

where $\mathcal{L}_A^{\text{geo}} = \sqrt{\|D_A^z - D_{A_{priv}}^{\text{geo}}\|_F^2}$ and $D_A^z$ denote the Euclidean distances in the view A private latent space, and geometric losses for other latent spaces are defined analogously. $\lambda_{\text{geo}}$ is a hyperparameter balancing the original disentanglement and reconstruction objectives with the new geometry preservation objective. The geodesic distances ($D^{\text{geo}}$) are estimated once prior to fine-tuning, using the trained network from Step 1. The latent space Euclidean distances ($D^z$, $D^s$) are recomputed at each fine-tuning epoch as the encoder parameters $\theta_{ae}$ are updated (Algorithm 2). This encourages the encoders to find mappings that reflect the data's shared and private submanifold structure, providing more interpretable representations.

## 3 RESULTS

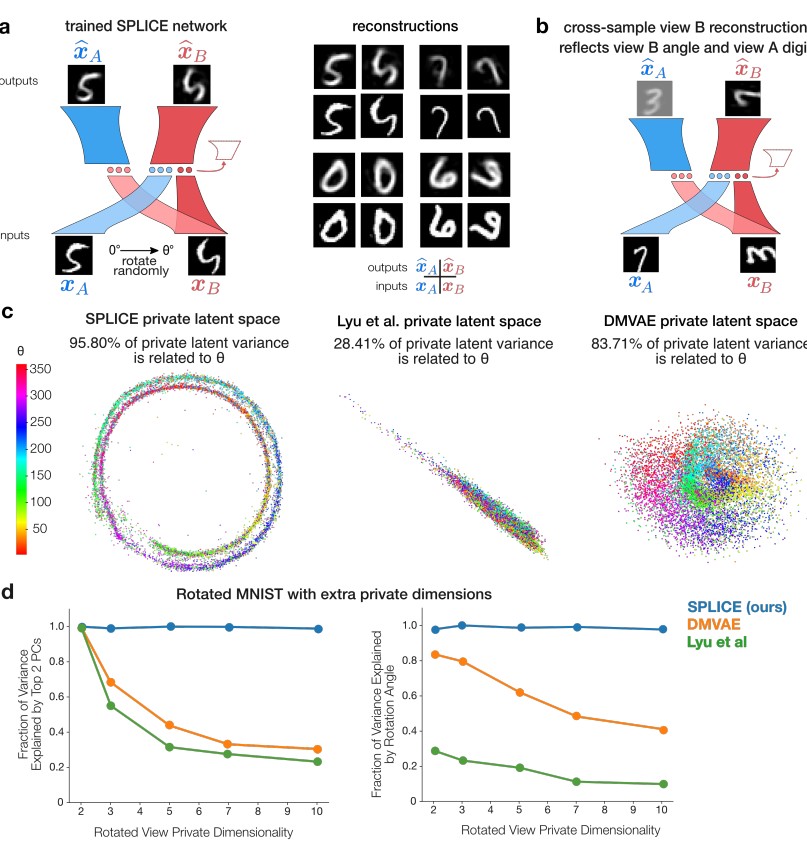

Figure 2: Rotated MNIST example. **a)** (left) During training, view $A$ inputs were original MNIST digits, view $B$ inputs were a random rotation of them. (right) SPLICE accurately reconstructs the original and rotated digits. **b**: The $F_B(\cdot)$ private encoder in SPLICE distilled from input $x_B$ only the rotation angle and discards digit identity, indicating successful disentangling. **c)** SPLICE retains the circular 1-D geometry of rotation angle, unlike Lyu et al. (Lyu et al., 2021) and DMVAE (Lee & Pavlovic, 2021). **d)** Even when given 5x the true number of private latents, SPLICE confines private variance to 2 dimensions, while other methods use all available dimensions and admit non-angle related variance.

**Experiment 1: Rotated MNIST.** We first validated SPLICE on rotated MNIST digits, a common dataset in multi-view learning. The dataset consisted of paired samples of an original MNIST image (view A) and randomly rotated versions of the same image (view B). To better distinguish the capabilities of SPLICE and existing methods, we uniformly sampled the angle of rotation from a full circle ($\theta \in [0°, 360°]$), rather than the limited range ($-45°$ to $45°$) typically used in the multi-view

learning literature (Wang et al., 2015; Lyu et al., 2021; Karakasis & Sidiropoulos, 2023). The shared information in this experiment was thus the digit identity and features (e.g., line thickness), while the view $B$ private information was the rotation angle. Since the $x_A$ inputs were not rotated, there was no private information for $A$.

After training, SPLICE accurately reconstructed the original and rotated digits (Fig. 2a). The inferred private latent space $\hat{z}_B$ was a 1D double circular manifold, along which rotation angle steadily increased (Fig. 2c). The proximity of angles 180◦ apart makes sense considering that vertically symmetric digits look almost identical when rotated 180◦, and even non-symmetric digits activate similar pixels when rotated 180◦. Both shared latent space $\hat{s}_{A\to B}$ and $\hat{s}_{B\to A}$ showed clear organization by digit, with clusters for similar digits (e.g. 4 and 9, 3 and 8) closer together (Supp. Fig. 3), and no apparent representation of angle (Supp. Fig. 7.

We compared SPLICE's performance to two private-shared disentangling methods: Lyu et al. (2021) and Lee & Pavlovic (2021) (DMVAE). Lyu et al. (2021) uses an deterministic Deep-CCA-based architecture, with an adversarial scheme to disentangle shared and private latents. DMVAE uses a variational autoencoder framework, with a total correlation-based disentangling objective and a mixture-of-experts inference for cross-modality generation. Lyu et al. (2021) produced a latent space with no clear angular organization (Fig. 2c, middle), which we suspect resulted from leakage between shared and private information. While DMVAE (Fig. 2c, right) does show visible organization by angle, it failed to extract a 1D circular manifold, and instead produced a circular point cloud, presumably due to its isotropic Gaussian prior on the latent space. These discrepancies illustrate the fundamental limitations of existing methods for blind scientific discovery. If we did not know beforehand that the true private latent was the angle of rotation, only SPLICE's inferred latent space would have provided clues that the private latent variable was a 1D circular variable. Lyu et al. (2021)'s latent space would instead suggest a 1D linear variable, and DMVAE's would suggest two largely independent 1D linear variables.

We quantified SPLICE's performance relative to other methods by calculating the amount of private latent variance explained by the true angle of rotation. In line with the qualitative results, rotation angle explained a greater proportion of the variance in SPLICE's private latent space compared to those of the other two methods (Fig. 2c; Supp. Fig. 5c, Supp. Table 1)), indicating that SPLICE achieves better disentangling than existing multi-view methods. To assess the fidelity of SPLICE's shared latent space, we additionally assessed how well digit identity could be decoded from the SPLICE shared latent space, relative to Lyu et al. (2021), Lee & Pavlovic (2021), and three shared-only methods: Andrew et al. (2013), Wang et al. (2015) and Karakasis & Sidiropoulos (2023). The decoding accuracy from SPLICE's shared latent space was close to the best competing method (Supp. Fig. 3, Supp. Table 5), and the latent space showed no apparent organization by rotation angle, the private latent variable (Supp. Fig. 7).

To assess the robustness of SPLICE and existing private-shared methods to mis-specified latent dimensionality, we also trained versions of each model with more private latent dimensions than necessary. Importantly, SPLICE confined virtually all private latent variance to two dimensions – even when given 5x the required number of dimensions – indicating that SPLICE is robust to mis-specified latent dimensionality (Fig. 2d). In contrast, Lyu et al. and DMVAE had considerable latent variance in the extra dimensions and considerable latent variance unrelated to rotation angle (Fig. 2d). These results highlight another crucial advantage of SPLICE over existing methods for blind discovery; if we had no a priori knowledge of the true latent dimensionality and picked a dimensionality that was too large, only SPLICE would have suggested that the true private latent was confined to a 2D plane.

The disentangled SPLICE latents significantly generalized, allowing us to generate arbitrarily rotated digits for digit-angle combinations not in the training set. Interestingly, we were able to verify that the new projections do lie on the original data manifold (Supp. Fig. 5b). The Lyu et al. model was unable to compose digit identity and angles from different test samples (Supp. Fig 5a), and DMVAE was largely successful but sometimes applied an incorrect rotation angle (Supp. Fig. 5a). Finally, we also trained SPLICE with different values for $\lambda_{geo}$ and found that SPLICE is remarkably robust to this hyperparameter . The private and shared latent spaces were quantitatively and qualitatively similar even for order-of-magnitude differences of $\lambda_{geo}$ (Supp. Table 2, Supp. Fig 6).

Finally, to assess whether our disentangling and geometry preservation approach generalized to different classes of encoder/decoder architectures, we trained a SPLICE model with convolutional

encoders and decoders on the rotated MNIST dataset. The disentangling quality and geometry of the learned private latents were similar to the fully connected version (Supp. Fig. 11, which indicates that our approach is somewhat agnostic to specific architectural details. We found similar results and performance improvement over previous disentangling methods on a different rotated images data set, composed of synthetically-generated "sprites" (Supp. Section A.3).

**Experiment 2: Synthetic LGN-V1 activity**.

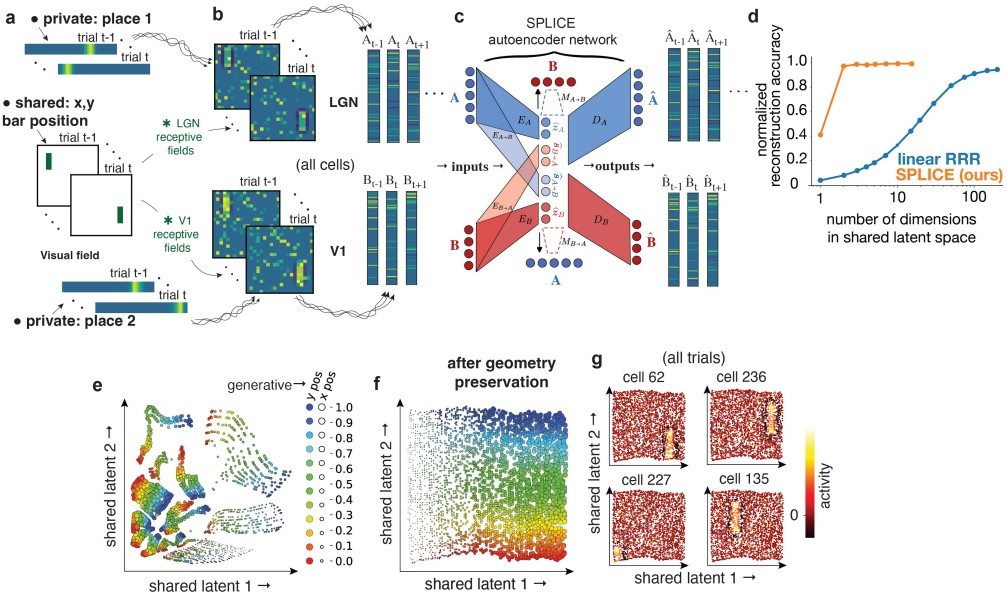

Figure 3: Simulated LGN-V1 experiment. **a)** The two synthetic brain regions encode 1) location on a linear track (place 1, private to $A$), 2) the 2D position of a vertical visual stimulus bar (shared across $A$ and $B$), and 3) a second linear track location (place 2, private to $B$). **b)** The visual stimulus drives center-surround and Gabor RFs. Neurons are ordered by RF centers; As the neurons' private place infromation is unrelated to their visual RF centers, it appears as random noise. **c)** SPLICE autoencoder network. **d)** SPLICE correctly estimates the shared latent dimensionality ($d = 2$), while RRR overestimates it as $d = 70$. **e)** Value of the 2D SPLICE shared latents for each trial (each dot is one trial) before applying geometry preservation. **f)** Same as g, but after applying SPLICE's geometry preservation. $x$ and $y$ positions are dot size and dot color, respectively. **g)** Each panel shows the data as in f, but colored by the activity of a randomly chosen neuron: SPLICE allows discovering that the activity coordinated across the regions has localized RFs that tile the shared space.

Motivated by applications in neuroscience, we next probed SPLICE's ability to handle nonlinear, neural-like data. We simulated two populations of neurons responding to a shared "visual" stimulus – a rectangular bar with variable x and y positions – and private 1D "position" stimuli (Fig. 3a). To mimic neuronal receptive fields, the first population had lateral geniculate nucleus (LGN)-like center-surround receptive fields to the visual stimulus, and the second population had V1-like Gabor-filter receptive fields to the visual stimulus (Supp. Fig. 8). Both populations had Gaussian tuning curves to their respective private stimuli, mimicking the tuning curves of hippocampal place field cells. To make this dataset more challenging, we set the variance of the shared response to be $\approx$ 6x smaller than the variance of the private response.

After training, SPLICE accurately reconstructed the simulated neural responses (Fig. 3c, right). The full SPLICE model correctly recovered a 2D sheet in the shared latent space, with the axes corresponding to the true x and y coordinates of the visual stimulus (Fig. 3f). Both inferred private latent spaces captured the corresponding ground truth "position" variable, showing near perfect correlation between ground truth and inferred latents (Supp. Figs 9,10). Remarkably, we were able to estimate the receptive fields for each simulated neuron by plotting a heatmap of their activity in the inferred latent space – a key capability for blind neuroscientific discovery (Fig. 3g).

To highlight the importance of SPLICE's geometry preservation step, we examined the shared latent space for an ablated model that was trained with only the Step 1 loss. The resulting latent space was highly fragmented and distorted, but still organized by stimulus x and y position (Fig. 3e). If we had no access to the ground truth shared latents, the ablated model would suggest that the shared information had a complex fragmented structure. However, using the full model, while still blind to the ground truth, would allow us to correctly infer that the true shared information consisted of 2 independent linear variables organized as a simple, continuous 2D sheet.

We first compared SPLICE's performance on this dataset to Reduced Rank Regression (Semedo et al., 2019), a popular linear method in neuroscience for identifying shared latent variables. A common paradigm in neuroscience is to estimate the true latent dimensionality of neural activity by gradually increasing dimensionality and identifying when reconstruction quality saturates. Following the same approach, we found that SPLICE correctly identified the true shared dimensionality ($n_s = 2$; Fig. 3c, orange). In contrast, RRR required $\approx$75 dimensions for reconstruction to saturate (Fig. 3c, blue). Because we designed our simulated neurons to have the same tuning curves as real neural populations, the discrepancy between SPLICE and RRR confirms the importance of nonlinear models in gaining a clear description of shared variablility between populations; RRR grossly overestimated the number of shared dimensions due to its linear assumptions.

Similarly to the ablated SPLICE model, competing nonlinear methods from the machine learning literature produced latent spaces that were highly fragmented (Supp. Figs. 9,10). To quantify the interpretablity of the latent spaces, we calculated how well the ground truth latents could be decoded linearly from the latent spaces. We found that SPLICE was able to decode the shared and private ground truth latents nearly perfectly ($R^2 > 0.99$), while competing methods had poor decoding accuracy.

Finally, we assessed the ability of SPLICE to disentangle shared and private in the presence of noise. We added i.i.d. Gaussian noise to the neuron responses, and found that SPLICE was able to recover the shared and private geometry even when the variance of the i.i.d. noise was 0.4 times the variance of the signal, resulting in a shared SNR of 0.329 and a private SNR ratio of 2.166. (Supp. Fig. 8d,e).

**Experiment 3: Data from neurophysiological experiments:** Having shown SPLICE's advantages in disentangling and blind discovery on neural and non-neural synthetic datasets, we turned to showing SPLICE's utility on experimental neurophysiological data. Specifically, we wanted to assess whether SPLICE could blindly rediscover shared information between regions that is known from the neuroscience literature. We fit SPLICE to electrophysiologically-recorded neural data from simultaneous Neuropixels recordings of hippocampus and prefrontal cortex, taken as mice performed a decision making task in a Virtual Reality T-maze (Fig. 4a). In single-region studies, both these brain regions have been shown to encode the animal's spatial position.

SPLICE's inferred shared latent space showed a shared encoding of the animal's position (Fig. 4c), consistent with the presence of place cells in both brain regions. Indeed, we could reliably decode the animal's position from the shared latent space on held-out trials ($R^2 = 0.889$). SPLICE also outperformed Reduced Rank Regression (Semedo et al., 2019) in that reconstruction quality saturated with just two dimensions, while RRR required $\approx$ 12 for reconstruction to saturate (Fig. 4b). This discrepancy suggests that, like in Experiment 2, the relationship between shared information and neural responses is nonlinear, and the linear model was unable to correctly distill the shared information into a small number of latent dimensions.

## 4 DISCUSSION

We propose SPLICE as an unsupervised approach for learning interpretable latent representations of shared and private information in complex, high-dimensional paired data sets. Compared to existing methods (Semedo et al., 2019; Lyu et al., 2021; Lee & Pavlovic, 2021), SPLICE more effectively disentangles shared and private information, yields more interpretable latents by preserving geometry, is more robust to incorrect estimates of latent dimensionality. While we preserve submanifold geometry using L-Isomap (Silva & Tenenbaum, 2002) due to its relative simplicity and computational efficiency, the SPLICE framework supports any manifold learning technique that produces pairwise geodesic distances, e.g., robust extensions of Isomap (Budninskiy et al., 2018), diffusion-based distances (Moon et al., 2019), or dynamics-based distances (Low et al., 2018).

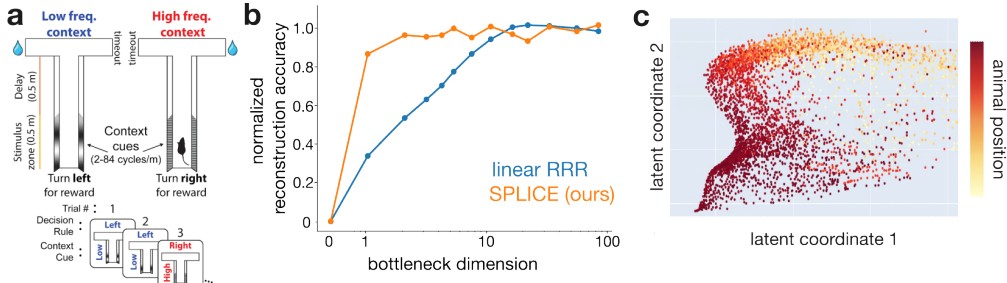

Figure 4: Neurophysiological data experiment. **a)** On each trial, mice made a Left/Right decision on a virtual T-maze. The correct response was cued by a visual stimulus in the first half of the stem of the T. Simultaneous recordings yielded 96 neurons in hippocampus, and 348 neurons in medial prefrontal cortex. **b)** SPLICE outperformed RRR (Semedo et al., 2019), summarizing the shared space in 2 dimensions, instead of 12. **c)** The shared latent space encodes the animal's position.

A key design choice in SPLICE is disentangling via predictability minimization, which explicitly encourages "first-order independence" ($E[x_B|z_A] = $ constant) for continuous-valued latents. While a stronger condition than decorrelation, this condition is theoretically weaker than the full independence condition of Lyu et al. (2021). Empirically, however, our experiments show that Lyu et al.'s method disentangles less effectively than methods with weaker theoretical guarantees, consistent with previous evaluations (Karakasis & Sidiropoulos, 2023). Thus while Lyu et al.'s guarantees hold in idealized conditions, practical implementations face challenges with limited network size and imperfect optimization, which require balancing theoretical guarantees with empirical performance. Future work could explore loss functions that better balance this trade-off. Nevertheless, SPLICE's Step 1 alone achieved superior disentangling than leading methods across diverse datasets (Supp. Tables 1,3,4), indicating that first-order independence is effective for learning disentangled representations.

**Hyperparameter tuning:** For SPLICE, we tuned the batch size, learning rate, weight decay, and network architecture, selecting the hyperparameters with the best cross-validated Step 1 objective function. The encoder and decoder architectures were constrained to always mirror each other, and the measurement network architectures were identical to the decoders. Some parameters did not require tuning; we used the same values of $n_{msr} = 5$ and $\lambda_{dis} = 1$ across all datasets, and set $\lambda_{geo}$ as the average data norm divided by the average geodesic distance. To ensure fair comparisons, we similarly selected hyperparameters for baseline methods via a consistent tuning procedure (details in Supplement). Although adversarial training can be brittle, we found that with the selected $\lambda_{dis} = 1$, SPLICE trained without issues (e.g. oscillatory behavior or divergence) for all datasets.

**Related work in machine learning:** To our knowledge, SPLICE is the only multi-view learning method that infers disentangled shared and private latent variables while preserving their submanifold geometry. Prior efforts focused on fully disentangling every dimension of every latent variable (Schmidhuber, 1992; Kim & Mnih, 2018; Geadah et al., 2018), which is an ill-posed unsupervised problem without further constraints (Locatello et al., 2020), and discards informative dependencies between latents. VAE approaches face similar issues due to their reliance on isotropic Gaussian priors (Kim & Mnih, 2018; Lee & Pavlovic, 2021; Palumbo et al., 2023), which encourage disentangling between each latent and are generally not appropriate for matching arbitrary data distributions. SPLICE instead only disentangles shared and private latent *spaces*, allowing the geometric loss to preserve structure within each space. Contemporaneous with our work, Kevrekidis et al. (2024) also use a crossed autoencoder structure to avoid private-to-shared information leakage, and enforce orthogonality constraints on encoder gradients to disentangle shared and private information. Unlike SPLICE, however, this method requires accurate prior estimation of the latent sizes to prevent shared-to-private information leakage, assumes that the private and shared submanifolds are orthogonal in the data space, and does not attempt to preserve latent submanifold geometry.

Although geometry-preserving loss terms have previously been used in autoencoder frameworks (Gropp et al., 2020; Lee et al., 2021), these methods are limited to single-view data and thus do not aim to disentangle groups of latent variables. Uscidda et al. (2024) attempts to obtain disentangled latent variables (using a isotropic Gaussian prior) and preserve Euclidean distances or

angles in single-view data, but these objectives are at odds when the distribution in observation space is not Gaussian, which is the case for most real data. Lederman & Talmon (2018) uses an alternating diffusion approach to learn shared latent manifolds from *multi-view data*, but alternating diffusion is conceptually unable to recover private latent geometry, since each set of private latents affects only one view. Furthermore, the method is non-parametric, and thus cannot quantify variance explained or easily embed new points outside the training set. These capabilities, which are enabled by SPLICE's autoencoder framework, are vital for assessing goodness of fit and analyzing new observations.

Like SPLICE, some previous multi-view methods can also learn latent representations without prior knowledge of the dimensionality. Gui et al. (2025) shows that CLIP (Radford et al., 2021), a multimodal foundation model, adapts to the intrinsic dimension of its training data. However, CLIP infers only shared latents, and is concerned with generation rather than the interpretability of the latents. Shrestha & Fu (2024) use sparsity-promoting objectives to learn statistically independent shared and private latent distributions without prior knowledge of dimensionality. However, to infer latents for individual samples, their GAN framework requires gradient-based inversion of the generator w.r.t its inputs (latents). This is quite slow for large numbers of samples; SPLICE's encoders explicitly learn the map from observation space to latent spaces, avoiding this limitation. Furthermore, neither of these methods attempt to preserve the shared and private submanifold geometry.

**Related work in neuroscience:** Within neuroscience, assessing the relationship between activity in different brain regions has relied on models that assume either a linear link between latents and neural activity (Semedo et al., 2019; Gokcen et al., 2022; 2023), or linear followed by a pointwise nonlinearity (e.g., $\text{softplus}(\cdot)$ or exponential) (Glaser et al., 2020; Balzani et al., 2022; Keeley et al., 2020; Gondur et al., 2023; Dowling & Savin, 2025). As artifical neural networks, all of these can be thought of as single-layer models between latents and neural activity. As neural representations are known to often be nonlinear, many datasets may not be well described with such approaches. Exceptions among prior work include (Abbaspourazad et al., 2024), which learns a highly nonlinear embedding of neural activity into a single latent state that evolves with a linear dynamical system, similar to Koopman operators (Koopman, 1931). While some of the studies implicity encourage disentangling of shared and private latents, there are no explicit disentangling terms, demonstrations of precise disentangling, nor explicit preservation of manifold geometry.

Another class of models in neuroscience learn relationships between multi-view data using either data augmentation and/or foundation-model architectures. Liu et al. (2021) introduces a self-supervised learning architecture which uses self- and cross-view reconstructions, and isotropic Gaussian latent priors to learn shared and private latent representations from augmentations of neural data. Their approach is highly similar to Lee & Pavlovic (2021) — the main difference being that Liu et al. (2021) has a weaker disentangling loss (only a VAE prior) than Lee & Pavlovic (2021) (which uses a total-correlation disentangling loss) — that we show that SPLICE outperforms. More recent foundation models (Liu et al., 2022; Chau et al., 2025) also aim to make cross-neuron or cross-modality predictions, but are concerned primarily with data prediction and generation. They thus use a single embedding space, implicitly learning the shared and private latent relationships within the black-box transformer networks. In contrast, SPLICE aims to enable interpretation of the shared and private latent variables, and thus focuses on learning explicitly disentangled representations that preserve submanifold geometry.

**Limitations:** A primary limitation of SPLICE is that it can only account for two views at once. Analyzing three or more would require multiple pairwise runs of SPLICE. This constraint is likewise faced by CCA and its extensions considered in this paper. A second drawback is that SPLICE targets only the geometry of the data, omitting information about temporal evolution. Future work should consider combining SPLICE with dynamical approaches that account for temporal structure. Lastly, SPLICE's assumes that 1) the joint shared-private latent distribution is the product of the shared and private block marginal distributions (i.e. shared and private latent spaces are statistically independent, a common assumption in the shared-private disentangling literature) and 2) observed data points sufficiently sample the joint shared-private latent distribution so that submanifold projections are valid. The second assumption follows from proper disentangling: if the inferred latent spaces from Step 1 are truly statistically independent, arbitrary combinations of the inferred shared and private variables will have a nearby point in the training data, and thus our projection step will not be extrapolating to unseen latent combinations. While this may not strictly hold for all real-world datasets, our MNIST experiment shows that the latent combinations not seen during training still lie on the data manifold, indicating tolerance to mild violations of this assumption.

## ETHICS STATEMENT

Our work provides a general model for obtaining interpretable descriptions of multi-view data. We focus on neuroscience applications, but the model could be applied to other settings with paired samples (e.g. sensor fusion, images from different viewpoints, etc). We do not foresee any negative societal impacts from our work.

## REPRODUCIBILITY STATEMENT

All models in this manuscript were train in PyTorch using the AdamW optimizer on an NVIDIA RTX 4080 GPU. Further details about hyperparameters and architecture for the experiments presented above are available in the Appendix. Upon publication, we will make available a GitHub repository containing the SPLICE implementation and scripts for running the experiments above.

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

# A APPENDIX

---

**Algorithm 1** Training process for Step 1, separating shared and private latents

---

1: Initialize autoencoder networks $\theta_{ae} = \{G_A, G_B, F_A, F_B, F_{A \to B}, F_{B \to A}\}$
2: Initialize measurement networks $M_{A \to B}$ and $M_{B \to A}$
3: **for** $i$ in $1 \ldots n_{iter}$ **do**
4:     Freeze encoders and decoders, unfreeze measurement networks
5:     **for** $iter$ in $1 \ldots n_{msr}$ **do**
6:         **for** $(A_{batch}, B_{batch}$ in $dataloader)$ **do**
7:             Use measurement networks to predict datasets from private latents: $x_A^{pred}$ and $x_B^{pred}$
8:             Compute measurement networks' prediction loss $\mathcal{L}_{pred}^A$ and $\mathcal{L}_{pred}^B$
9:             Update measurement networks to minimize $\mathcal{L}_{pred}^A + \mathcal{L}_{pred}^B$
10:         **end for**
11:     **end for**
12:
13:     Freeze measurement networks, unfreeze encoders and decoders
14:     **for** $(A_{batch}, B_{batch}$ in $dataloader)$ **do**
15:         Encode inputs $A_{batch}$ and $B_{batch}$ to get all latents: $\widehat{s}_{B \to A}$ and $\widehat{s}_{A \to B}$, $\widehat{z}_A$ and $\widehat{z}_B$
16:         Decode shared and private latents to reconstruct inputs: $\widehat{A}_{batch}$ and $\widehat{B}_{batch}$
17:         Compute reconstruction loss $\mathcal{L}_{rec}^A$ and $\mathcal{L}_{rec}^B$
18:         Update encoder and decoder networks to minimize $\mathcal{L}_{rec}^A + \mathcal{L}_{rec}^B$
19:
20:         Encode inputs $A_{batch}$ and $B_{batch}$ to get all latents: $\widehat{s}_{B \to A}$ and $\widehat{s}_{A \to B}$, $\widehat{z}_A$ and $\widehat{z}_B$
21:         Compute $\text{Var}[M_{B \to A}(\widehat{z}_B)]$ and $\text{Var}[M_{A \to B}(\widehat{z}_A)]$
22:         Update encoder networks to minimize $\text{Var}[M_{B \to A}(\widehat{z}_B)] + \text{Var}[M_{A \to B}(\widehat{z}_A)]$
23:     **end for**
24: **end for**

---

## A.1 MNIST EXPERIMENT

### A.1.1 RESULTS ACROSS MULTIPLE RANDOM SEEDS

Supplementary Table 1: MNIST disentangling results across multiple random seeds.

| | Var. in $z_b$ Exp. by $\theta$ | p-val. to SPLICE step 1 | p-val. to full SPLICE |
|---|---|---|---|
| Lyu et al. (2021) | $27.09 \pm 9.84$ | $p < 0.0001$ | $p < 0.0001$ |
| DMVAE | $84.64 \pm 1.35$ | $p = 0.0002$ | $p < 0.0001$ |
| SPLICE step 1 | $94.83 \pm 2.27$ | - | $p = 0.0262$ |
| SPLICE (both steps) | $\mathbf{97.12 \pm 0.78}$ | $p = 0.0262$ | - |

### A.1.2 RESULTS ACROSS MULTIPLE $\lambda_{geo}$ VALUES

Across orders of magnitude, different $\lambda_{geo}$ values produced similar distangling (below) and qualitatively similar latent spaces (Supp. Fig. 6).

Supplementary Table 2: MNIST disentangling results across multiple $\lambda_{geo}$ values

| $\lambda_{geo}$ | Var. in $z_b$ Exp. by $\theta$ |
|---|---|
| 0.001 | 98.28 |
| 0.01 | 97.12(mean) |
| 0.1 | 97.89 |

---

**Algorithm 2** Training process for Step 2, fine-tuning to preserve geometry

---

1: Select a random sample to generate $\widehat{z}_A^{\text{fix}}, \widehat{s}_{B \to A}^{\text{fix}}, \widehat{s}_{A \to B}^{\text{fix}}, \widehat{z}_B^{\text{fix}}$
2: Calculate submanifold projections $\widehat{x}_A^{z_A \text{ subm}}, \widehat{x}_A^{s_{B \to A} \text{ subm}}, \widehat{x}_B^{s_{A \to B} \text{ subm}}, \widehat{x}_B^{z_B \text{ subm}}$
3: Estimate geodesic distances $D_A^{\text{geo}}, D_{B \to A}^{\text{geo}}, D_{A \to B}^{\text{geo}}, D_B^{\text{geo}}$
4: **for** $i$ in $1 \ldots n_{iter}$ **do**
5:     Freeze encoders and decoders, unfreeze measurement networks
6:     **for** $iter$ in $1 \ldots n_{msr}$ **do**
7:         **for** $(A_{batch}, B_{batch}$ in $dataloader)$ **do**
8:             Use measurement networks to predict datasets from private latents: $x_A^{pred}$ and $x_B^{pred}$
9:             Compute measurement networks' prediction loss $\mathcal{L}_{pred}^A$ and $\mathcal{L}_{pred}^B$
10:            Update measurement networks to minimize $\mathcal{L}_{pred}^A + \mathcal{L}_{pred}^B$
11:         **end for**
12:     **end for**
13:
14:     Freeze measurement networks, unfreeze encoders and decoders
15:     **for** $(A_{batch}, B_{batch}$ in $dataloader)$ **do**
16:         Encode inputs $A_{batch}$ and $B_{batch}$ to get all latents: $\widehat{s}_{B \to A}$ and $\widehat{s}_{A \to B}$, $\widehat{z}_A$ and $\widehat{z}_B$
17:         Decode shared and private latents to reconstruct inputs: $\widehat{A}_{batch}$ and $\widehat{B}_{batch}$
18:         Compute reconstruction loss $\mathcal{L}_{rec}^A$ and $\mathcal{L}_{rec}^B$
19:         Compute geometry preservation loss $\mathcal{L}_{geo} = \mathcal{L}_{geo}^A + \mathcal{L}_{geo}^B + \mathcal{L}_{geo}^{A \to B} + \mathcal{L}_{geo}^{B \to A}$
20:         Update encoder and decoder networks to minimize $\mathcal{L}_{rec}^A + \mathcal{L}_{rec}^B + \lambda_{geo} \mathcal{L}_{geo}$
21:
22:         Encode inputs $A_{batch}$ and $B_{batch}$ to get all latents: $\widehat{s}_{B \to A}$ and $\widehat{s}_{A \to B}$, $\widehat{z}_A$ and $\widehat{z}_B$
23:         Compute $\text{Var}\left[M_{B \to A}(\widehat{z}_B)\right]$ and $\text{Var}\left[M_{A \to B}(\widehat{z}_A)\right]$
24:         Update encoder networks to minimize $\text{Var}\left[M_{B \to A}(\widehat{z}_B)\right]$ and $\text{Var}\left[M_{A \to B}(\widehat{z}_A)\right]$
25:     **end for**
26: **end for**

---

### A.1.3 MNIST DATASET DETAILS

For the MNIST experiment, we used the MNIST dataset Lecun et al. (1998) under the GNU General Public License v3.0. We first split the dataset into training, validation, and test sets, with 50000 digits in the training set, 10000% in the validation set, and 10000% in the test set. Therefore, the performance metrics we report on the MNIST dataset are based on held-out test set digits that were unseen during training *at any rotation*. Each paired data sample was generated by randomly selecting one of the digits in the corresponding datasplit, and applying rotations by a random angles $\theta \in [0°, 360°]$. View A was the original digit, and view B was the same digit rotated by $\theta$ degrees. We only included one rotation of each original MNIST digits, so the final dataset consisted of 50000 training samples, 10000 validation samples, and 10000 test samples.

### A.1.4 HYPERPARAMETER SELECTION FOR SPLICE AND BASELINES

The original papers for all models we compared to included an experiment with rotated MNIST digits. We therefore used the same hyperparameters as in the original papers and did not perform a hyperparameter search for this experiment.

For all models, we used 30 latent dimensions for each shared space, 0 dimensions for the unrotated view's private space, and 2 dimensions for the rotated view's private space. For SPLICE, we used a fully connected network with hidden layers of $[256, 128, 64, 32]$ for the encoders. The decoders mirrored the encoder architecture. The measurement networks had the same architecture as the decoders. We trained SPLICE for 100 epochs with a batch size of 100, learning rate of $10^{-3}$, and weight decay of $10^{-3}$. We set the coefficient for the disentangling loss to 1 for simplicity, and set the coefficient for the geometry preservation loss to be 0.01 based on the approximate ratio between the average geodesic distance and the average norm of the data. Training SPLICE took approximately 30 minutes on a single NVIDIA RTX4080 GPU. For SPLICE Step 2, we used 100 neighbors and 100 landmarks for the geodesic distance calculations. We chose the value of 100 neighbors by starting at

100 and increasing in increments of 100 until the nearest neighbors graph was not fragmented. For the other models, we used the same hyperparameters and hidden layer sizes as in the original papers.

For the convolutional version of SPLICE, the encoders consisted of two convolutional layers of stride 2 with 64, then 32 4x4 convolutional kernels, followed by two fully connected layers of 1568 and 256 units. The decoders were a mirrored version of the encoders, and the measurement networks were the same as in the fully connected version. This encoder/decoder architecture matches the original Lyu et al. (2021); Lee & Pavlovic (2021) papers.

### A.1.5   CORRECTING FOR INITIAL ROTATION IN VARIANCE EXPLAINED CALCULATION

We noted that the estimated angle was at times offset by the inherent angle of the digits in the MNIST dataset; Some of the digits were written at an angle before any rotations were applied. To correct for the baseline angle, we obtained the latent values for many rotations of each digit, which yielded a curve of the latent angle as a function of the rotation angle for each digit and model. We then identified for each digit a horizontal shift in the latent angle estimation by maximizing the inner product of each digit's angle curve with a reference curve selected from one random digit (Supp. Fig. 4). This offset was then subtracted to provide an absolute angle inclusive of the initial built-in angle.

We then calculated the variance explained in the private latents by the angle by selecting samples in 2 degree windows over the true "corrected" angle of the digits, calculating the variances of the corresponding private latents, and dividing by the total variance of the private latents. Subtracting this value from 1 gives the variance explained by the angle of the digits for each window. The final variance explained was calculated as the average of the variances in each window.

### A.1.6   ASSESSING ON-MANIFOLD PRESENCE OF GENERATED DIGITS

We quantified if these projections lie on the original data manifold by calculating the distances between the private submanifold (i.e. the arbitrarily rotated digits) and the nearest neighbors in the observed dataset, and repeated a similar calculation for the shared submanifold. For both submanifolds, the distributions of submanifold nearest-neighbors distances were similar to the distribution nearest-neighbors distances between observed data points. Virtually all projection nearest-neighbors distances were smaller than the average within-digit-class distance. These distance metrics suggest that the projections do lie on the original data manifold, despite not training the model with pairs consisting of different digits (Supp. Fig. 5b).

## A.2   LGN-V1 EXPERIMENT

### A.2.1   SIMULATION DETAILS

Given a stimulus consisting of a bar of light presented at different positions in different trials, datasets $A$ and $B$ are the activity of a field of simulated LGN neurons and V1 neurons, respectively. The stimuli were kept at a single orientation (vertical). By construction, the ground truth shared information across both views is the X and Y position of the bar, which geometrically is a 2-dimensional sheet. The LGN population consisted of 400 neurons, with center-surround receptive fields whose centers were evenly spaced on a two-dimensional 20x20 grid. The V1 population consisted of two evenly spaced 20x20 grids of neurons with Gabor filter receptive fields (i.e., V1 was 800-dimensional). The first grid had vertically oriented Gabor filters and the second had horizontally oriented Gabor filters. The visual field was implemented as a 100x100 pixel grid, and the size of each neuron's receptive field was 30x30 pixels (Supp. Fig.8a).

In addition to the shared visual stimulus, each population also responded to a private 1-D stimulus. For each population, this was generated by placing a virtual agent along a 1-D virtual linear track. Each neuron had a randomly centered Gaussian place field on this linear track. On different trials, the LGN agent and the V1 agent were placed at random, mutually independent, positions on the track. The neuronal responses to the shared and private stimuli were added linearly to obtain the final activity for each neuron. We scaled the variance of the responses to the private latents to be 6X the variance of responses to shared latents. Supp. Fig. 3a shows example stimuli and inputs to the SPLICE network in Supp. Fig. 3b show an example of the resulting LGN and V1 population activity. In some simulations, we also added i.i.d. noise to each individual neuron. For each simulation, we generated 18,900 trials, with the stimulus placed at a randomly chosen X and Y position for each trial.

64% of the trials were used for training, 16% for validation, and 20% for the testing results shown in Supp. Fig. 3.

### A.2.2 HYPERPARAMETER SELECTION FOR SPLICE AND BASELINES

We compared SPLICE to DeepCCA Andrew et al. (2013), DeepCCAE Wang et al. (2015), Karakasis Karakasis & Sidiropoulos (2023), Lyu et al. Lyu et al. (2021) and DMVAE Lee & Pavlovic (2021) for the LGN-V1 dataset. We used fully connected networks for all models, with the decoder architecture mirroring the encoder architecture. For hyperparameter tuning, we used the Ray Tune library Liaw et al. (2018) with the HyperOptSearch algorithm Bergstra et al. (2013) and the ASHAScheduler Li et al. (2020), and optimized with respect to the objective function for each model on the validation set.

The discrete search space was defined as follows:

- Learning rate: $\{10^{-2}, 10^{-3}, 10^{-4}, 10^{-5}\}$
- Weight decay: $\{0, 10^{-1}, 10^{-2}, 10^{-3}, 10^{-4}\}$
- Batch size: $\{1000, 2000, 5000, 12096\}$
- # of units per hidden layer: $\{50, 100, 200\}$
- # of hidden layers: $\{2, 3, 4, 5, 6\}$

The best performing hyperparameters from this search space were:

- SPLICE: Learning rate $10^{-3}$, weight decay $10^{-3}$, batch size 12096, # of units per hidden layer 200, # of hidden layers 6
- DeepCCA: Learning rate $10^{-3}$, weight decay 0, batch size 2000, # of units per hidden layer 200, # of hidden layers 3
- DeepCCAE: Learning rate $10^{-2}$, weight decay 0, batch size 5000, # of units per hidden layer 200, # of hidden layers 2
- Karakasis: Learning rate $10^{-3}$, weight decay 0, batch size 1000, # of units per hidden layer 200, # of hidden layers 3
- Lyu et al.: Learning rate $10^{-3}$, weight decay $10^{-4}$, batch size 1000, # of units per hidden layer 200, # of hidden layers 6
- DMVAE: Learning rate $10^{-3}$, weight decay $10^{-3}$, batch size 1000, # of units per hidden layer 200, # of hidden layers 6

Other hyperparameters were set according to the original papers (see Sprites section above). For SPLICE, we set the coefficient for the disentangling loss to 1 for simplicity, and set the coefficient for the geometry preservation loss to be 0.05 based on the approximate ratio between the average geodesic distance and the average norm of the data. For SPLICE Step 2, we used 200 neighbors and 100 landmarks for the geodesic distance calculations. We chose the value 200 by starting at 100 and increasing in increments of 100 until the nearest neighbors graph was not fragmented. SPLICE was trained for 25000 epochs, which took approximately 2.5 hours on a single NVIDIA RTX4080 GPU.

### A.3 SPRITES EXPERIMENT

### A.4 SPRITES EXPERIMENT

### A.4.1 SPRITES RESULTS

We also assessed SPLICE's ability to disentangle on simple synthetic data with known true private and shared information. Our dataset consisted of images of "sprites" (Fig. 1a), each defined by a specific configuration of features (hair, pants, shirt, etc.). For each sample, we selected a single sprite and rotated it by random angles $\theta_A, \theta_B \in [0°, 360°]$ to produce $x_A$ and $x_B$ (Fig. 1b). Thus, the *shared* information was the sprite identity (i.e., the sprite features) and the *private* information was the view-specific rotation angle. The private submanifolds of this dataset, each corresponding to all

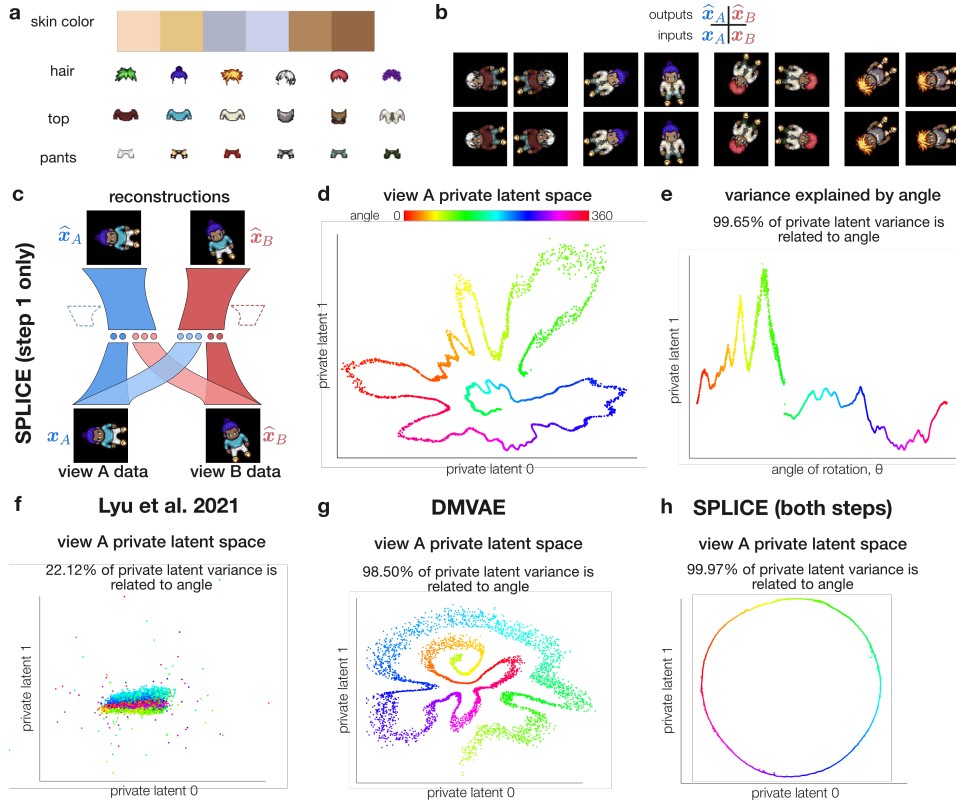

Supplementary Figure 1: Sprites dataset. **a)** Attributes used to generate unique sprites. **b)** Each paired sample consisted of a single sprite rotated by random angles $\theta_A$ and $\theta_B$. **c-d)** SPLICE Step 1 found a private latent space $z_A$ tightly organized by the rotation angle, indicating excellent disentangling. **e)** Lyu et al. found a latent space organized by angle along only one dimension. **f)** DMVAE found a latent space organized by angle, but with poorer disentangling than SPLICE. **g)** Applying SPLICE Step 2 to the network from **c-d)** produced a private latent space with a nearly perfect ring geometry.

possible rotations of a single sprite, were thus 1D circular manifolds (i.e. rings) due to the periodic nature of $\theta$.

We trained SPLICE Step 1 with 500-dimensional shared latents and 2-dimensional private latents on this dataset. After training, the network successfully reconstructed its inputs (Supp. Fig. 1b), explaining $95.93\%$ of the variance in $x_A$ and $96.19\%$ in $x_B$. The private latent space $\hat{z}_A$ was organized by $\theta_A$, indicating that the network successfully distilled only the rotation angle into the private latent (Supp. Fig. 1c,d). Indeed, $\theta$ accounts for $99.65\%$ of the total variance in $\hat{z}_A$, indicating a high degree of disentangling. Similarly, the sprite identity accounted for $99.75\%$ of the total variance in $\hat{s}_{B \to A}$. Compared to Lyu et al. (Lyu et al., 2021) and DMVAE (Lee & Pavlovic, 2021), SPLICE Step 1 explained more data variance and achieved better disentangling for all latent spaces (Supp. Table 4, Supp. Table 3).

Although the true geometry of $\theta_A$ is a ring, nonlinear networks in SPLICE Step 1 and the other methods obscure this geometry by cutting and warping the latent space $\hat{z}_A$ (Supp. Fig. 1c-f), highlighting the need for the geometry preservation Step 2 of SPLICE. Applying Step 2 to the Step 1-trained network produced private latent spaces that still encoded rotation angles, but had geometries that were nearly perfect rings (Supp. Fig. 1g). Thus, if we did not know beforehand that the angle was the true private information (as is the case for unsupervised discovery), SPLICE's discovery of the ring geometry would have provided the insight that the private information was a 1-D circular variable. Looking at the Step 1 latent space would not have yielded such insight in this scenario.

Supplementary Table 3: SPLICE disentangling and reconstruction vs. baselines on Sprites dataset

| | reconstruction $R^2$ | | SD Exp. by $\theta_i$ (%) | | SD Exp. by sprites(%) | |
|---|---|---|---|---|---|---|
| | $\widehat{\boldsymbol{x}}_A$ | $\widehat{\boldsymbol{x}}_B$ | $\widehat{\boldsymbol{z}}_A$ | $\widehat{\boldsymbol{z}}_B$ | $\widehat{\boldsymbol{s}}_A$ | $\widehat{\boldsymbol{s}}_B$ |
| Lyu et al. | 0.904 | 0.895 | 5.67 | 16.28 | 3.51 | 3.95 |
| DMVAE | 0.885 | 0.882 | 88.11 | 87.09 | 94.06 | 93.97 |
| SPLICE (Step 1) | **0.959** | **0.962** | 94.15 | 93.31 | 95.58 | 95.19 |
| SPLICE (full) | 0.956 | 0.953 | **98.83** | **98.91** | **98.43** | **98.44** |

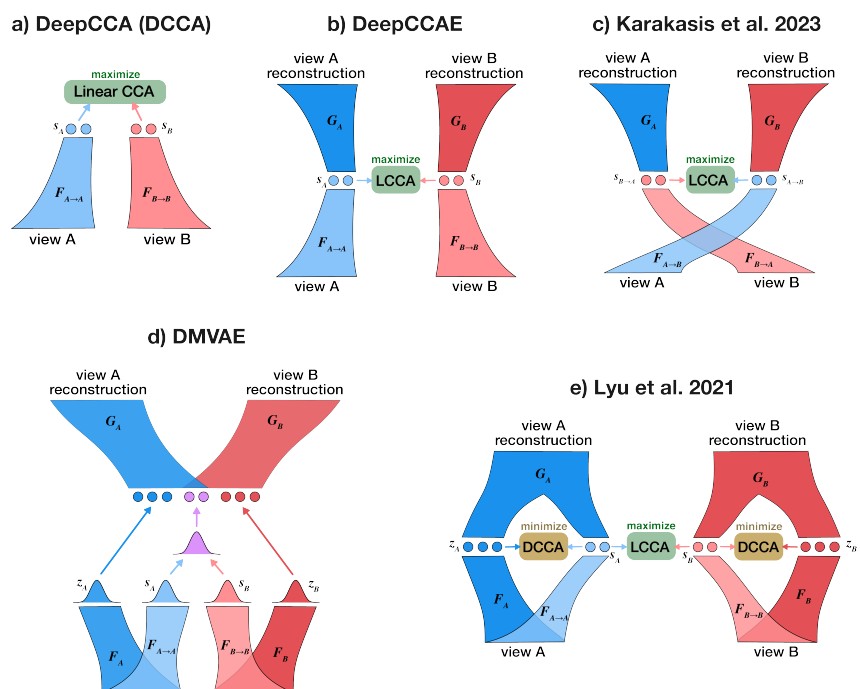

Supplementary Figure 2: **a)** DeepCCA infers shared latent variables by extending linear CCA to use deep neural network encoders. **b)** DeepCCAE attempts to capture as much shared information as possible by adding decoders to DeepCCA. **c)** Karakasis et al. Karakasis & Sidiropoulos (2023) crosses the encoders in DCCAE to eliminate leakage of private information into the shared latents. **d)** Lyu et al. Lyu et al. (2021) explicitly models private latent variables, and uses adversarial DeepCCA networks to encourage disentangling between shared and private latents. **e)** DMVAE Lee & Pavlovic (2021) explicitly models private latents in a variational framework and encourages disentangling through minimizing the total correlation – the KL divergence between the joint latent distribution and the product of the marginal latent distributions.

### A.4.2 SPRITES DATASET DETAILS

For the Sprites experiment, we used static sprite frames from Li & Mandt (2018), available at `https://github.com/YingzhenLi/Sprites` under a CC-BY-NC 4.0 license. The dataset contains animations of 2D sprites, each with a unique combination of skin color, hair, top, and pants. Each attribute had 6 possible values, for a total of $6^4 = 1296$ unique sprites.

For our experiment, we used only the first frame of each animation. Importantly, we first split the unique sprites into training, validation, and test sets, with 80% of the sprites in the training set, 10% in the validation set, and 10% in the test set. Therefore, the performance metrics we report on the Sprites dataset are based on held-out test set sprites that were unseen during training *at any rotation*.Each

Supplementary Table 4: SPLICE disentangling vs. baselines on Sprites dataset

| | Var. Exp. by $\theta_i$ (%) | | Var. Exp. by Sprite ID (%) | |
| --- | --- | --- | --- | --- |
| | $\widehat{z}_A$ | $\widehat{z}_B$ | $\widehat{s}_A$ | $\widehat{s}_B$ |
| Lyu et al. | 10.97 | 22.12 | 6.76 | 7.70 |
| DMVAE | 98.50 | 98.31 | 99.09 | 98.97 |
| SPLICE (step 1 only) | 99.65 | 99.55 | 99.79 | 99.75 |
| SPLICE (both steps) | **99.99** | **99.99** | **99.97** | **99.97** |

paired data sample was generated by randomly selecting one of the sprites in the corresponding datasplit, and applying rotations by random angles $\theta_A, \theta_B \in [0°, 360°]$. The final dataset consisted of 20000 training samples, 5000 validation samples, and 5000 test samples.

### A.4.3 Tuning procedure a for SPLICE and baselines

We compared SPLICE to Lyu et al. Lyu et al. (2021) and DMVAE Lee & Pavlovic (2021) for the Sprites dataset. We used fully connected networks for all models, with the decoder architecture mirroring the encoder architecture. For hyperparameter tuning, we used the Ray Tune library Liaw et al. (2018) with the HyperOptSearch algorithm Bergstra et al. (2013) and the ASHAScheduler Li et al. (2020), and optimized with respect to the objective function for each model on the validation set.

The discrete search space was defined as follows:

- Learning rate: $\{10^{-2}, 10^{-3}, 10^{-4}, 10^{-5}\}$
- Weight decay: $\{0, 10^{-1}, 10^{-2}, 10^{-3}, 10^{-4}\}$
- Batch size: $\{500, 1000, 2000\}$
- Hidden layer sizes:
    - $[1024, 512, 512, 2048]$
    - $[1024, 512, 512, 2048, 1024]$
    - $[1024, 512, 512, 2048, 1024, 512]$

The best performing hyperparameters from this search space were the same across all models: Learning rate $10^{-4}$, weight decay $10^{-3}$, batch size 1000, and hidden layer sizes $[1024, 512, 512, 2048, 1024, 512]$. Because we wanted to select hyperparameters in a completely unsupervised manner, we did not use the validation set to select hyperparameters that affected the calculation of the objective functions, i.e. coefficients for loss terms. We instead set these coefficients to the same values recommended (see below) in the original papers.

### A.4.4 Model architectures and training for SPLICE and baselines

For all models, we used 500 latent dimensions for each shared space and 2 latent dimensions for each private space. Measurement networks for SPLICE had the same architecture as the decoder networks, and the DCCA networks for Lyu et al. consisted of 3 fully connected layers with 64 hidden units each, as suggested in the original paper. All models were trained for 5000 total epochs.

Additional hyperparameters were set as follows:

- Lyu et al. Lyu et al. (2021): $lr_{max} = 1$, $decay_{mmcca} = 10^{-1}$, $\beta = 1$, $\lambda = 100$
- DMVAE Lee & Pavlovic (2021): $\lambda = 10$, $\beta = 1$
- SPLICE: $\lambda_{disent} = 1$, $\lambda_{geo} = 0.005$.

The values for Lyu et al. were selected as recommended in the original paper. For DMVAE, the paper and code provided conflicting values for the coefficients, so we contacted the authors and set the

values per their recommendation. For Lyu et al., $lr_{max}$ and $decay_{mmcca}$ are the learning rate and weight decay for the adversarial DCCA networks, $\beta$ is the coefficient for the reconstruction loss, and $\lambda$ is the coefficient for the disentangling loss. For DMVAE, $\lambda$ is the coefficient for the reconstruction loss, and $\beta$ is the coefficient for the total correlation term of the KL expression.

For SPLICE, we set the coefficient for the disentangling loss to 1 for simplicity, and set the coefficient for the geometry preservation loss to be 0.05 based on the approximate ratio between the average geodesic distance and the average norm of the data. For SPLICE Step 2, we used 500 neighbors and 100 landmarks for the geodesic distance calculations. We chose the value 500 by starting at 100 and increasing in increments of 100 until the nearest neighbors graph was not fragmented.

### A.4.5 VARIANCE EXPLAINED CALCULATION

We calculated the variance explained in the private latents by the angle by selecting samples in 2 degree windows over the true angle, calculating the variances of the corresponding private latents, and dividing by the total variance of the private latents. Subtracting this value from 1 gives the variance explained by the angle of the digits for each window. The final variance explained was calculated as the average of the variances in each window.

For the variance explained in the shared latents by the sprite ID, we calculated the variance of the shared latents for each unique sprite ID, and divided by the total variance of the shared latents. Subtracting this value from 1 gives the variance explained by the sprite ID for each window. Because our dataset had multiple different rotations for each sprite ID, this gave us enough repetitions of each sprite ID to obtain a good estimate of the variance explained by the sprite ID. The final variance explained was calculated as the average of the variances for each sprite ID.

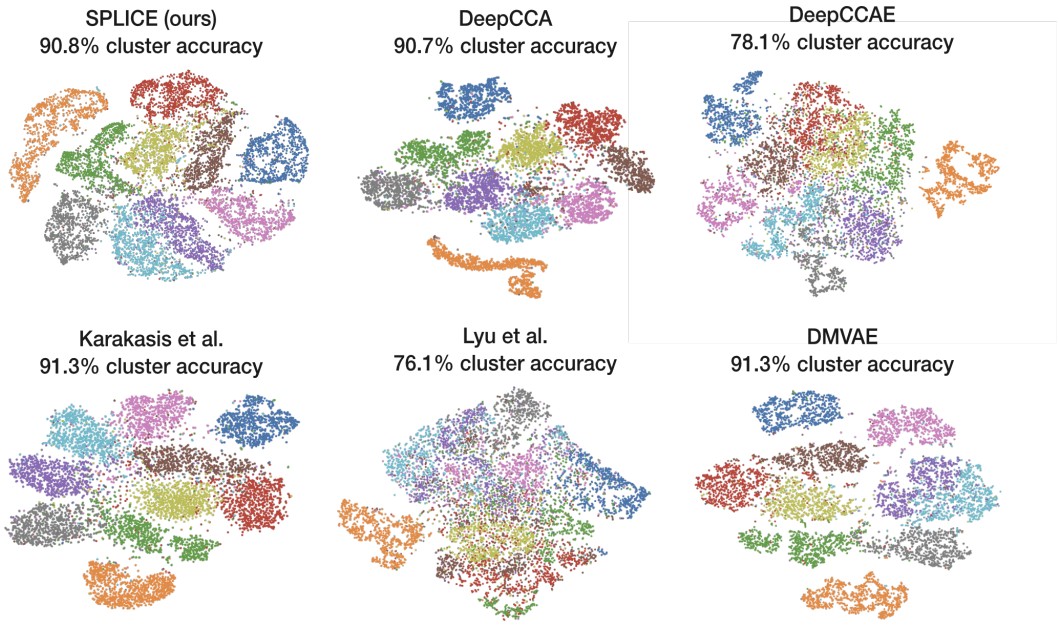

Supplementary Figure 3: tSNE visualization of clusters in the shared latent spaces that SPLICE successfully separates digits into distinct clusters. Competing methods tend to have less clear clusters (e.g., DeepCCAE and Lyu et al.) and at best comparable clustering (DeepCCA, Karkasis et al., and DMVAE) as measured by the classification accuracy. We note that clustering accuracy is an imperfect measure of shared info, as it does not account for other aspects of digit identity, such as digit style, line thickness, etc.

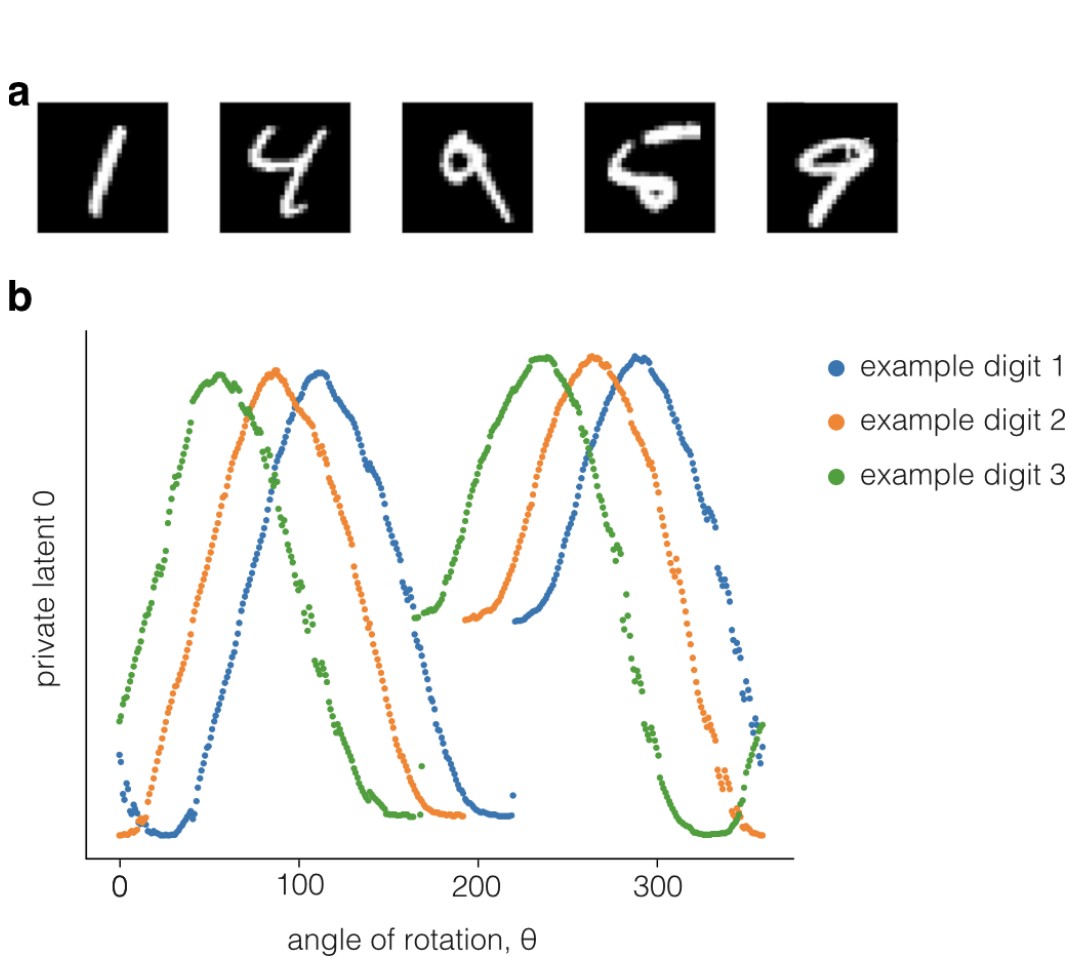

Supplementary Figure 4: **a)** MNIST digits have initial rotations – all digits shown are nominally unrotated. **b)** Example rotation vs. latent curves that were aligned to correct for inital rotation.

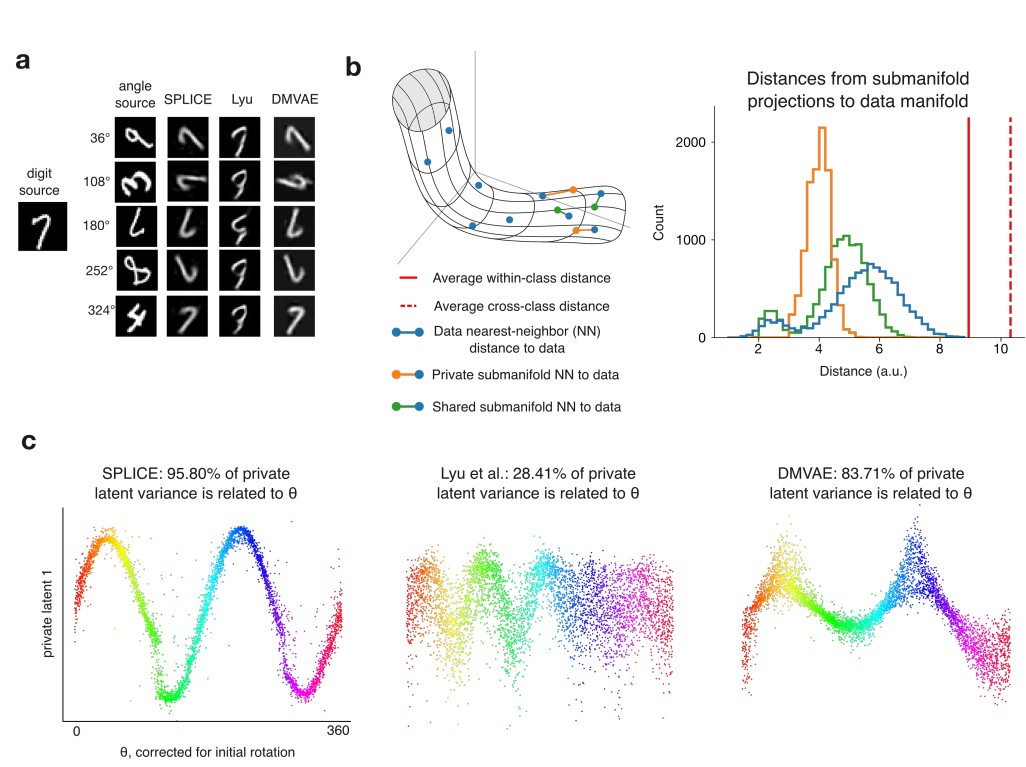

Supplementary Figure 5: **a)** Example cross-reconstructed digits. **b)** SPLICE cross-reconstructed digits lie on the data manifold. **c)** SPLICE obtains more disentangled private latents than competing methods.

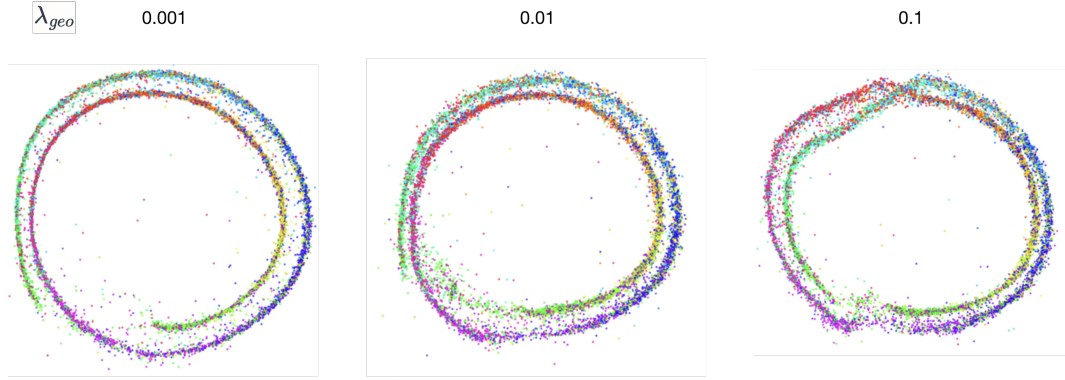

Supplementary Figure 6: SPLICE is robust to the choice of $\lambda_{geo}$. MNIST private latent spaces look qualitatively similar across 3 orders of magnitude of $\lambda_{geo}$.

Supplementary Table 5: SPLICE performance vs. baselines on MNIST dataset

|  | Shared Lat. Clustering Acc. (%) | Private Lat. Var. Exp. by $\theta$ (%) |
| --- | --- | --- |
| DeepCCA | 90.7 | – |
| DeepCCAE | 78.1 | – |
| Karakasis | 91.3 | – |
| Lyu et al. | 76.1 | 28.41 |
| DMVAE | 91.3 | 83.71 |
| SPLICE | 90.8 | 95.80 |

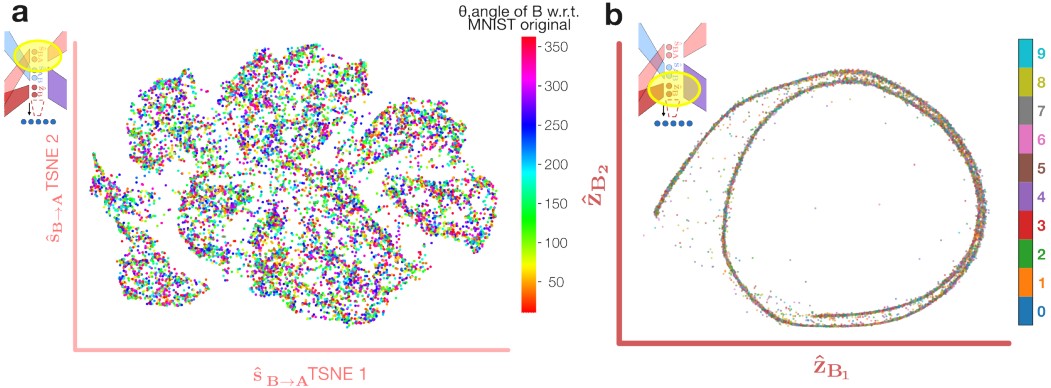

Supplementary Figure 7: SPLICE latent spaces show no apparent contamination by the opposite information type

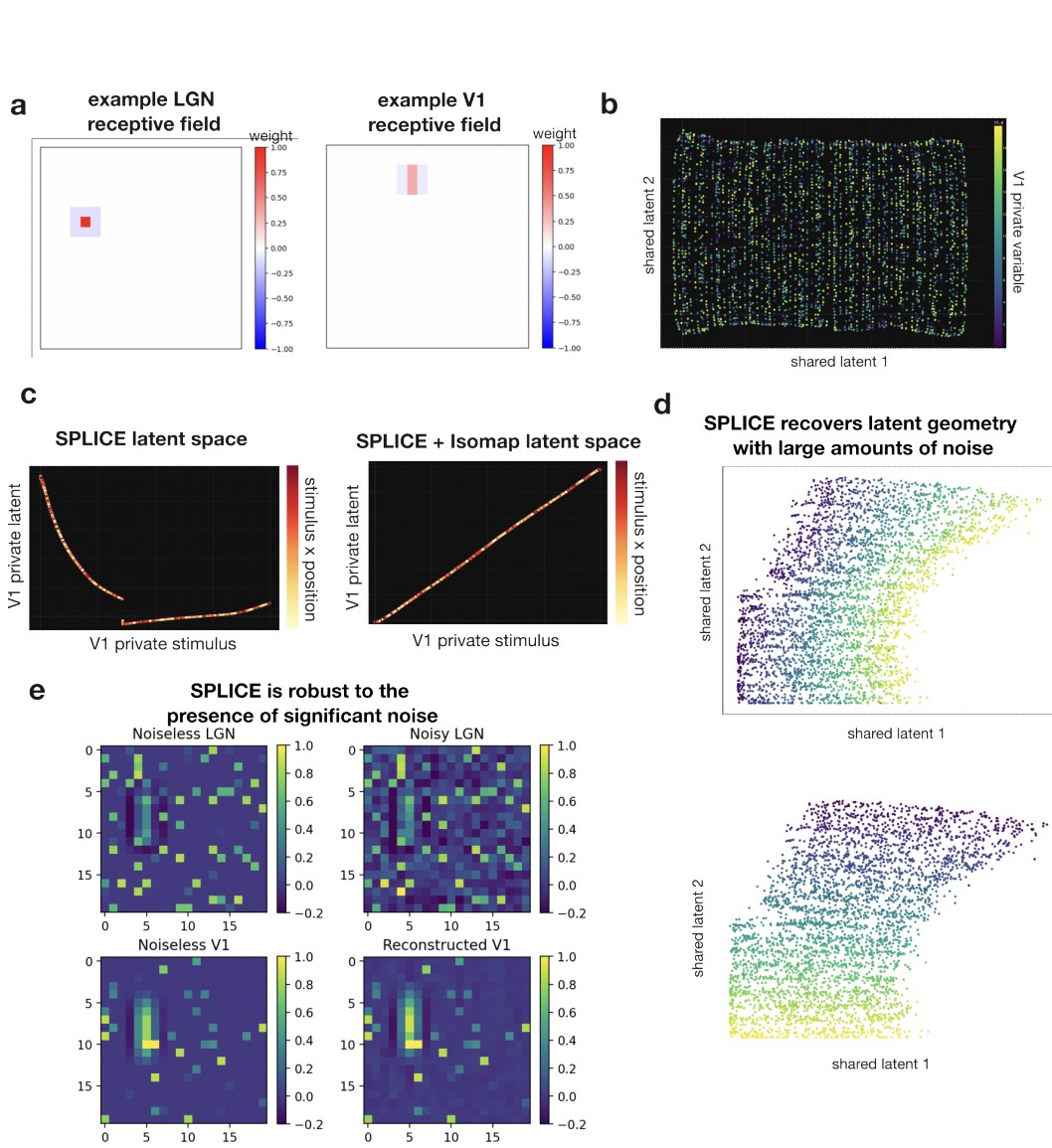

Supplementary Figure 8: LGN-V1 extras: **a)** Example receptive fields for simulated LGN and V1 neurons. **b,c)** SPLICE latent spaces are highly disentangled; coloring the shared latent space by ground truth private latents and vice versa shows no clear structure. **d,e)** SPLICE recovers the shared latent geometry and reconstructs well even in the presence of large amounts of i.i.d. noise (shown panels are when the variance of the noise is 0.4x the variance of the signal).

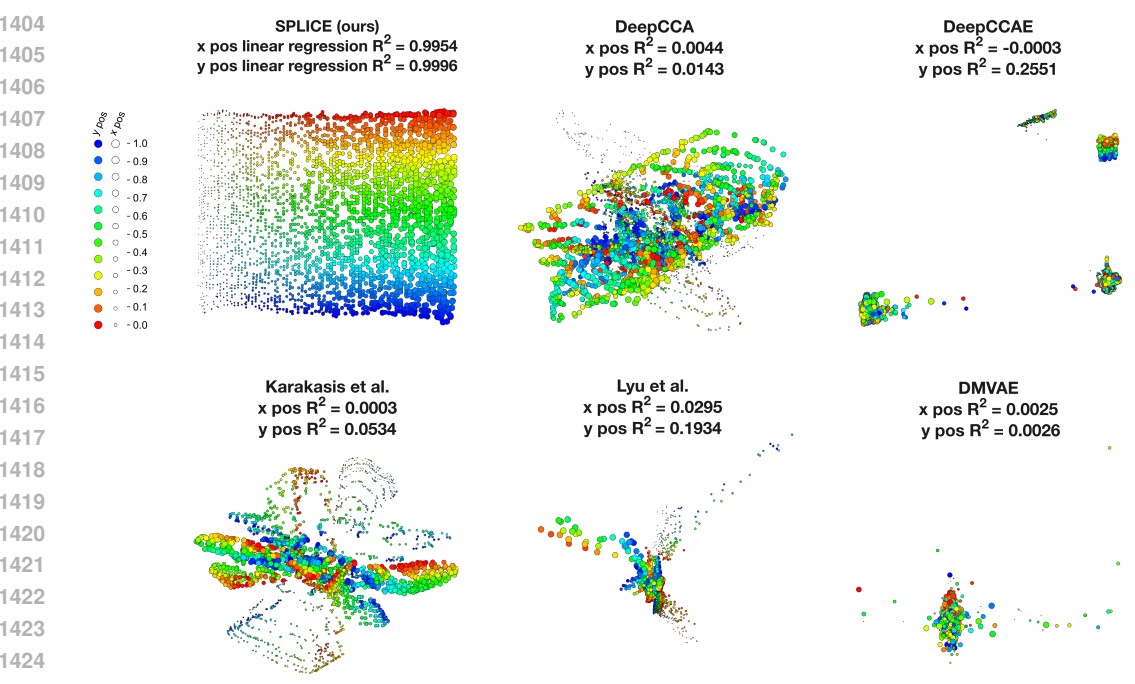

Supplementary Figure 9: LGN-V1 shared latent space: True vs. inferred private latents for methods that estimate shared latents. SPLICE substantially outperforms the competing methods in latent estimation, recovering a 2D sheet organized by stimulus X and Y position.

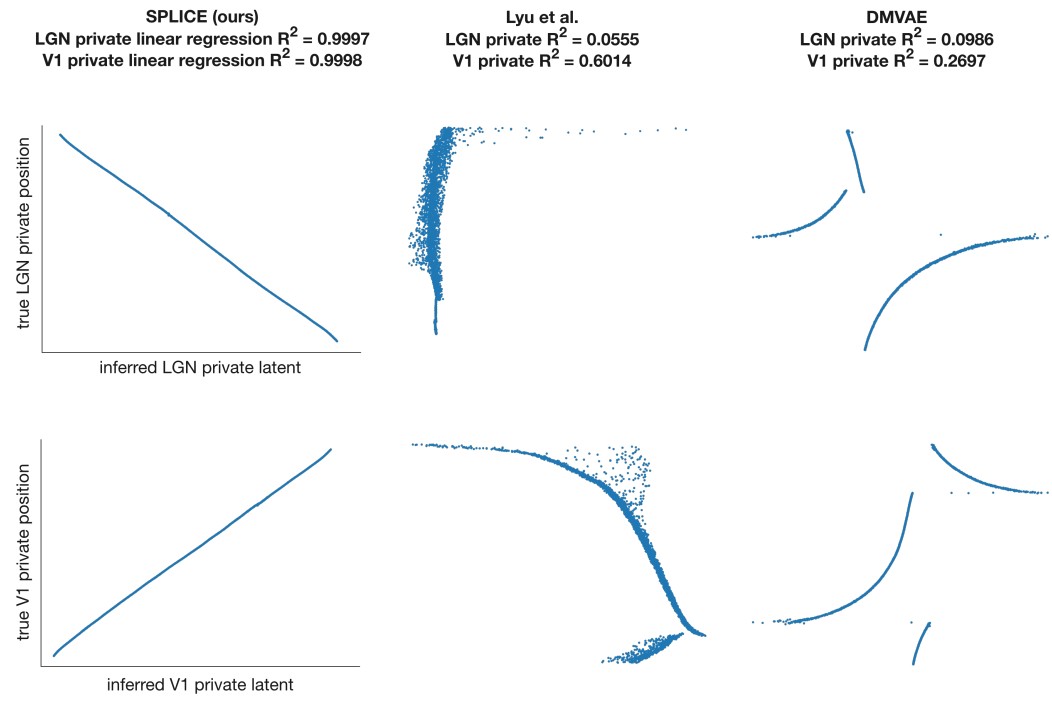

Supplementary Figure 10: LGN-V1 private latent space: True vs. inferred private latents for methods that estimate private latents. SPLICE substantially outperforms the competing methods (despite all models achieving good reconstruction quality), obtaining highly disentangled 1D structure in the private latents.

**SPLICE w/ conv. nets (both steps)**

97.75% of private latent variance
is related to θ

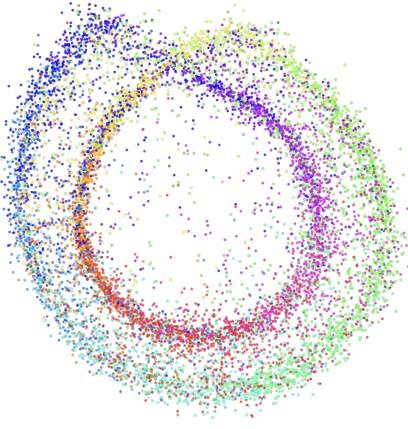

Supplementary Figure 11: Rotated MNIST private latent space for a SPLICE model trained with convolutional encoders and decoders. The variance explained by rotation angle and qualitative geometry are similar to the fully connected SPLICE model, suggesting that SPLICE's disentangling and geometry preservation loss terms are effective for multiple different classes of network architectures.

