# OpenReview forum: "Unsupervised discovery of the shared and private geometry in multi-view data"
_ICLR.cc/2026/Conference — Submitted to ICLR 2026_

### Official Review · Reviewer_UzvA · 2025-10-26

**Soundness:** 4
**Presentation:** 4
**Contribution:** 4
**Rating:** 6
**Confidence:** 4

**Summary:**

This paper introduces SPLICE, a novel framework for learning disentangled and interpretable representations of shared and private latent variables from paired observations of high-dimensional data.

This is a fundamental problem with implications across multiple domains, from neuroscience to multimodal representation learning. The authors provide a conceptually elegant and theoretically motivated approach grounded in predictability minimization and manifold geometry preservation. They validate SPLICE on three complementary scenarios, demonstrating substantially superior performance to prior methods.

The paper is well-structured, methodologically coherent, and convincingly analyzed, offering both theoretical insight and practical utility.

**Strengths:**

The use of predictability minimization for enforcing first-order independence between shared and private latents is original and intuitively appealing, addressing information leakage issues in earlier methods.

The experiments are carefully designed to illustrate disentangling, interpretability, and robustness to latent dimensionality mis-specification.
The application to real neural data adds strong credibility and relevance.

Beyond neuroscience, the framework provides a general solution for analyzing multiview high-dimensional data, potentially benefiting multiple research communities.

**Weaknesses:**

1. While SPLICE is presented as a general framework, the actual performance is closely tied to specific neural architectures used for encoding, decoding, and measurement. It remains unclear whether the proposed disentangling principles will generalize across architectures or scales without extensive hyperparameter tuning. A discussion or an analysis of architectural sensitivity would strengthen the claims.

2. While the approach is conceptually elegant, some arguments (e.g., the assumption that the full manifold is the cross-product of submanifolds) remain heuristic rather than theoretically justified.

3. Preserving submanifold geometry improves interpretability but may come at a computational or robustness cost, particularly for noisy or large-scale data.

**Questions:**

1. The proposed method relies on different neural architectures to learn different subproblems (shared, private, measurement). Given this dependence, to what extent can the theoretical ideas meaningfully guide the design of simpler or more scalable architectures?

2. Given the adversarial training component, what specific stabilization strategies were most effective in preventing mode collapse?

---

> ### Author Response · Authors · 2025-11-24
> **Response to Reviewer UzvA**
>
> **Specific Architectures:** We have added results to the supplement from a SPLICE model fit to the MNIST experiment using convolutional rather than fully connected networks and the same hyperparameters otherwise (Supp. Fig. 11). The disentangling quality and geometry of the learned representations are similar to the fully connected version, which indicates that our approach is somewhat agnostic to specific architectural details.
>
> **Weakness 2:** We agree with the reviewer that our paper focuses on the empirical advantages of our method over existing approaches, and does not include thorough theoretical justifications or guarantees. Our use of the term cross-product was confusing; we merely meant that we assume that the joint shared-private latent distribution can be factorized into the product of the marginal distributions of the shared and private latent spaces – this is the definition of statistical independence and is likewise assumed by most other shared-private disentangling methods. The only additional assumption necessary for our projection step is that the observed data points sample the joint shared-private latent distribution adequately enough that the decoders do not have to extrapolate the learned latent->observation mapping to unseen parts of the joint latent space. We have updated the text of our Limitations sections to more clearly state these assumptions.
>
> **Computational Complexity:** We agree with the reviewer that our submanifold geometry preservation requires additional complexity compared to purely disentangling methods, but we feel that especially for scientific machine learning which seeks to draw insights from the learned representations, the interpretability from geometry preservation is worth the additional computational costs. With regard to noise, we also fit SPLICE to a version of the LGN-V1 synthetic dataset with added i.i.d. Gaussian noise and found that SPLICE was indeed able to identify the submanifolds even with considerable noise magnitudes (Supp. Fig. 8). Our method can also be made more robust to noise by swapping L-Isomap out for a more noise-robust estimation of geodesic distances.
>
> **Question 1:** We note that for each experiment, we constrained the encoders, decoders, and measurement networks to have the same network architectures. To ensure that the measurement networks can properly detect and remove leakage of shared info into private latents, an important architectural constraint is that the measurement networks must be at least as expressive as the encoder/decoder networks.
>
> **Adversarial Training Stability:** The primary stabilization strategy for the adversarial training was ensuring that the measurement networks were always well-trained. We accomplished this by using measurement networks with identical architectures to the encoder and decoder networks (rather than shallower or less expressive networks), and by training the measurement networks for multiple iterations for each iteration of encoder/decoder training.
>
> While overtraining the discriminator is not always sufficient for traditional adversarial schemes (e.g. GANs), this seems to be sufficient in our case because the adversarial disentangling objective is an auxiliary objective in addition to the main reconstruction objective (There is some evidence that including autoencoding reconstruction losses in a GAN training scheme can prevent mode collapse and stabilize training [1]). In practice, this was enough to avoid brittleness in SPLICE’s adversarial training – we did not observe any cycling or mode collapse across multiple random seeds for all of our experiments. This is corroborated by the very low standard errors computed for the MNIST example over multiple random seeds (Table 1).
>
> [1] M. Rosca, B. Lakshminarayanan, D. Warde-Farley, and S. Mohamed, “Variational Approaches for Auto-Encoding Generative Adversarial Networks,” Oct. 21, 2017, arXiv: arXiv:1706.04987. doi: 10.48550/arXiv.1706.04987.

---

> > ### Comment · Reviewer_UzvA · 2025-11-26
> >
> > Thanks for your clarifications.

---

### Official Review · Reviewer_R5mr · 2025-11-01

**Soundness:** 3
**Presentation:** 3
**Contribution:** 3
**Rating:** 6
**Confidence:** 3

**Summary:**

This paper presents a method called SPLICE, which uses private and shared latent space to produce disentangled and interpretable representations. The proposed method also contains geometry identification and preservation components, which increases the fidelity of the latent space, leading to interpretable intrinsic sub-manifolds representations that preserves geometry. The proposed method is validated on controlled simulation datasets, and also is applied on neural datasets with different brain regions.

**Strengths:**

- The authors study the important problem of private/shared information disentanglement, which is a highly important problem in computational neuroscience.
- The proposed geometry identification and preservation method is novel, and very relevant in the neuroscience domain. The produced latent geometry can help study small-scale datasets, potentially help with scientific discoveries.
- The experimental designs are very detailed. Key ablations such as latent dimensions are carefully studied.

**Weaknesses:**

- This work [1] seems highly relevant, which also proposes a private/shared latent space disentanglement method for neuroscience datasets. The authors should benchmark this work with the proposed work. The authors should also consider benchmarking with more advanced transformer architectures, such as [2, 3].

[1] Liu, Ran, Mehdi Azabou, Max Dabagia, Chi-Heng Lin, Mohammad Gheshlaghi Azar, Keith Hengen, Michal Valko, and Eva Dyer. "Drop, swap, and generate: A self-supervised approach for generating neural activity." Advances in neural information processing systems 34 (2021): 10587-10599.

[2] Liu, Ran, Mehdi Azabou, Max Dabagia, Jingyun Xiao, and Eva Dyer. "Seeing the forest and the tree: Building representations of both individual and collective dynamics with transformers." Advances in neural information processing systems 35 (2022): 2377-2391.

[3] Chau, Geeling, Christopher Wang, Sabera Talukder, Vighnesh Subramaniam, Saraswati Soedarmadji, Yisong Yue, Boris Katz, and Andrei Barbu. "Population transformer: Learning population-level representations of neural activity." ArXiv (2025): arXiv-2406.

- The presented experiments are very small scale. The authors should consider include results from larger datasets.

**Questions:**

NA

---

> ### Author Response · Authors · 2025-11-24
> **Response to Reviewer R5mr**
>
> **Related work:** We appreciate the pointers to the additional relevant work. We note that the focus of these papers is to develop general purpose models, rather than analyze single datasets. That said, the architecture and disentangling approach used in [1] are extremely similar to the DMVAE method that we already compare to. Although [1] focuses on paired augmentations of a single view and DMVAE focuses on multi-view data, both methods use a variational autoencoder approach and incorporate loss functions that feature self- and cross-view reconstruction (i.e. with swapped shared latents). Importantly, DMVAE uses an additional total-correlation based loss to encourage statistical independence between latent spaces, while the method in [1] relies (implicitly) on the VAE gaussian prior on the latents for disentangling. Therefore, [1] is arguably a weaker disentangling baseline than DMVAE. [2,3] are also very interesting. However, as they aim to solve a different problem (prediction), they do not explicitly disentangle the shared and private information, which is key to interpreting the latent representations for blind scientific discovery. We have added a paragraph to our discussion section explaining the relationship of SPLICE to these related methods.
>
> **Small data:** Although the reviewer is correct that our datasets are small-scale in the context of the broader machine learning literature, our target application in developing SPLICE was in-vivo recordings of neural populations. For this data class, the application to electrophysiology measurements using Neuropixels probes represents data on the higher end of the dimensionality we can expect.  Our synthetic datasets were also designed to have equal or higher-dimensional observations than the real Neuropixels data. We agree that it would be very interesting to think of more foundation-model style learning, however we believe that this is outside the scope of the work as it would focus more on generic latent representations rather than the specific geometry for a given experimental dataset.

---

### Official Review · Reviewer_P8Fy · 2025-11-03

**Soundness:** 4
**Presentation:** 4
**Contribution:** 3
**Rating:** 8
**Confidence:** 4

**Summary:**

They propose a new deep learning method to reveal shared latent variables and separate latent variables across two datasets (or "views"). The architecture consists in two non-linear auto-encoders which swap parts of their latent space. In order to ensure that the private part of the latent does not leak shared information, they adapt an adversarial strategy proposed by (Schmidhuber 1992), predictability minimization, where a decoder aims at predicting the other dataset, while the global objective is to minimize the performance of that decoder.

**Strengths:**

- I find the method original yet simple, and well justified conceptually. I appreciate the elegant adaptation of the predictability minimization scheme to this problem.
- The 3 experiments demonstrating the quality of the method and superiority over other methods (DMVAE and non-linear CCA) are fitting and convincing. The experiments are also nicely complementary, going from toy datasets to real neuroscience problems.
- The writing is very clear for the most part (and some of my questions may have more to do with my own limited expertise than with clarity).
- The results presented in experiments 2 and 3 are quite promising from the point of view of scientific discovery in neuroscience, and I believe that this architecture has the potential to be useful in many scientific applications beyond neuroscience.

**Weaknesses:**

1. "Using shared latents from one view to reconstruct the other view guarantees that private information does not leak into the shared latents" => Not sure of that because the shared latent could as well contain other information discarded by the decoder. Please tone down claim if my assessment is correct.
2. See Questions about robustness of the method (may not be weaknesses if properly addressed)
3. Remark: Reading the intro I was missing many citations such as reduced-ranked regression (RRR), CCA, non-linear CCA such as Barlow Twins, CKA, SSL, equivariant SSL, but then found most of them in Discussion. It would be better to announce in Intro that a lot of references will be found in discussion, and at least cite RRR and CCA in intro (ideally applied to neuroscience datasets).

**Questions:**

1. On robustness (I): how many hyperparameters and how finetuned they need to be to the specific datatset? This information does not come through from reading the paper.
2. On robustness (II): "We use these measurement networks in an adversarial disentangling scheme: in predicting the opposite region’s observations as well as possible, the measurement networks try to exploit any shared information that has leaked into the private latents." => This requires careful implementation in my experience to avoid cyclic behaviors. Did you experience these cyclic behaviors? How did you mitigate them? Is your solution robust?
3. I did not understand the rational for step 2 of the method (in particular Projecting onto Submanifolds). Please explain the rational more concisely.
4. "Importantly, SPLICE confined virtually all private latent variance to two dimensions" => what regularization is responsible for that?
5. LGN - V1 experiment: unclear to me what insights could we have gained from SPLICE here?


typo:
-zˆA  sometimes has the small hat sometimes the big hat

---

> ### Author Response · Authors · 2025-11-24
> **Response to Reviewer P8Fy (1 of 2)**
>
> **Intro citations**: We thank the reviewer for pointing this out. We now cite the original RRR and CCA papers and example applications to neuroscience in the introduction, and mention that more related work will be found in the Discussion.
>
> **Weakness 1 (cross autoencoder guarantee)**: We thank the reviewer for pointing this out. We meant that crossing the encoders prevents the shared latents used to reconstruct a view from containing private information about that view – we have revised the text to make this clear. The reviewer is correct that this could still result in private information about the other view leaking into the shared latents if the info is ignored by the decoder. In practice, we use weight decay when training our networks, and thus the encoders should not introduce additional information into the latents if it is not useful for minimizing one of the other objectives (e.g. reconstruction), which is consistent with SPLICE’s excellent empirical disentangling performance.
>
> **Question 1 (hyperparameters)**: For each experiment, we used the same network architectures for the encoders, decoders, and measurement networks. Thus, the hyperparameters for SPLICE are the basic network architecture, the disentangling coefficient $\lambda_{dis}$, the geometry preservation coefficient $\lambda_{geo}$, and the number of landmarks and neighbors used for submanifold geodesic estimation (inherited from L-Isomap). In Supp. Fig. 6, we show that on the MNIST experiment, SPLICE achieves similar disentangling quality and latent space geometry across a two-orders-of-magnitude range of $\lambda_{geo}$. This indicates that our framework is largely robust to the exact value of this hyperparameter. Sensitivities to the number of landmarks/neighbors are inherited from L-Isomap, but our method can be made more robust to these parameters by swapping out L-Isomap for more robust variants. Network architectures can be tuned via cross-validation on reconstruction loss. We used the same number of measurement network iterations (5) for all datasets, so we do not consider this as a hyperparameter that needs to be tuned. We have added a new paragraph to our discussion that describes which hyperparameters were tuned and which were held constant across datasets.
>
> **Question 2 (adversarial training robustness)**: The primary stabilization strategy for the adversarial training was ensuring that the measurement networks were always well-trained. We accomplished this by 1) using measurement networks with identical architectures to the encoder and decoder networks (rather than shallower or less expressive networks) and 2) training the measurement networks for multiple iterations for each iteration of encoder/decoder training. Together, these strategies should strongly encourage the measurement networks to find any information leakage caused by the encoder networks.
>
> While overtraining the critic is not always sufficient to stabilize traditional adversarial schemes (e.g. GANs), we believe that this is sufficient in our case because the disentangling objective, which we use the adversarial scheme for, is an auxiliary objective (in addition to the reconstruction/geometry preservation) rather than the sole objective. Specifically, there is some evidence that including autoencoding reconstruction losses in an adversarial training scheme can prevent mode collapse and stabilize training [1]. In practice, this seemed sufficient to avoid brittleness in SPLICE’s adversarial training – we did not observe any cycling or mode collapse across multiple random seeds for all of our experiments. This is corroborated by the very low standard errors computed for the MNIST example over multiple random seeds (Table 1).
>
> [1] M. Rosca, B. Lakshminarayanan, D. Warde-Farley, and S. Mohamed, “Variational Approaches for Auto-Encoding Generative Adversarial Networks,” Oct. 21, 2017, arXiv: arXiv:1706.04987. doi: 10.48550/arXiv.1706.04987.

---

> > ### Author Response · Authors · 2025-11-24
> > **Response to Reviewer P8Fy (2 of 2)**
> >
> > **Question 3 (Step 2 Rationale):** The rationale for the two-step approach is that shared and private geometries are nonlinearly mixed in the observation space. Directly applying manifold learning techniques will not yield the shared and private submanifolds, since geodesic distances estimated from the original data points will mix private and shared variance. To estimate the geometry of the shared and private submanifolds via traditional manifold learning techniques, we must first remove the effect of the irrelevant submanifold’s information by projecting the data onto the submanifold of interest. We do this by fixing the set of latents associated with the irrelevant submanifold and varying the set of latents associated with the submanifold of interest, and mapping these new latent combinations to observation space via the decoder. Only given proper disentangling will the geodesic distances computed using these projections properly correspond to the geometry of the desired submanifold. We have updated the text in our Methods section to make this idea more clear.
> >
> > **Question 4 (Robustness to extra dimensions):** The disentangling loss is mainly responsible for SPLICE’s robustness to excess latent dimensions – specifically, the fact that we disentangle from private latents to the opposite view observations, rather than from private latents to shared latents. If the shared latents do not fully capture all the shared information (e.g. due to misspecified dimensionality), any extra shared information that leaks into the excess private dimensions will still be informative about the opposite view observations despite not being present in the inferred shared latents. Therefore, our adversarial disentangling scheme will push this information out of the private latents. Other disentangling approaches, which all (to our knowledge) disentangle between the inferred shared and private *latent spaces*, would not remove this extra shared info from the private latents. After the disentangling loss removes non-private information from the private latents, the geometry-preserving loss then ensures that the inferred private latents are not warped outside a 2D plane, since latent space Euclidean distances must match pairwise geodesic distances.
> >
> > **Question 5 (SPLICE insights from LGN-V1):** Compared to Reduced Rank Regression which grossly overestimated the latent space dimensionality, SPLICE affords the ability to visualize the latent space since it correctly infers a 2D latent space (reconstruction does not improve when we provide more than 2 latent dimensions). Compared to nonlinear methods which fragment the sheet (Supp. Fig. 9 & 10), SPLICE reveals that the underlying variability lies on a 2D sheet for shared and continuous 1D track for privates. If we did not know the ground truth latents beforehand, examining the latent spaces produced by competing nonlinear methods would suggest clustered or highly complex shared latents. Notably, SPLICE also lets us infer the receptive field of individual neurons on the latent sheet (Fig. 3g).
> >
> > We thank the reviewer for bringing the typos to our attention, we will correct those in the text.

---

> > > ### Comment · Reviewer_P8Fy · 2025-11-25
> > >
> > > I am satisfied with the author's answers to my questions and maintain my positive score.

---

### Official Review · Reviewer_NoRJ · 2025-11-05

**Soundness:** 3
**Presentation:** 2
**Contribution:** 1
**Rating:** 2
**Confidence:** 3

**Summary:**

This paper introduces a two-stage approach for unsupervised discovery of geometric structures in shared and private latent variables in multi-view data. The first stage uses an autoencoder while optimize for cross-latent (un-)predictability to induce disentanglement of shared and private latent variables. The second stage uses manifold learning technique to preserve geometric structures of data in latent space. Experiments are provided for rotated MNIST dataset, as well as synthetic and real data from neuroscience.

**Strengths:**

The paper is well written and easy to follow. The experiments showcase some synthetic and real scenarios where shared/private latent geometry is important but overlooked by the considered baselines, where the SPLICE outperforms. An application of the method in neural decoding is shown.

**Weaknesses:**

1. My main concern lies in novelty of the paper. While the applications to shared-private latent modeling in multi-view settings might be new, the cross-reconstruction framework used in Step 1 uses well-known technique for the purpose of aligning/disentangling latent representations (Schmidhuber, 1992; Chen et al., 2021). The manifold learning technique in Step 2 is from existing literature (L-ISOMAP), and isometry is also a widely adopted prior for representation learning in existing literature.


2. Missing discussions and experiments with related works:
- An advantage of SPLICE is to retain geometric structure of data, by encouraging isometry between data space and latent space. This seems similar to an existing line of work on geometric structure preservation in disentanglement, e.g. (Gropp et al., 2020, Lee et al., 2022, Uscidda et al., 2025). However, there is no discussion or comparison of SPLICE with these methods.
- Another advantage of SPLICE is to disentangle shared/private latents "without a priori knowledge of latent dimensionality" (line 81-83). Related works on this topic are missing, e.g., (Gui et al., 2025) showed that multi-modal contrastive learning adapts to intrinsic dimension of shared latent variables, or (Shrestha et al., 2025) where the authors aimed to tackle content-style learning with unknown latent dimensionality.


3. While qualitative results are provided for Experiment 2, an extensive quantitative result (e.g., with R^2 metric) was not provided.

**References**

Schmidhuber, “Learning Factorial Codes by Predictability Minimization”, Neural Computation, 1992.

Chen et al., “Exploring Simple Siamese Representation Learning”, CVPR, 2021.

Gropp et al., “Isometric Autoencoders”, arXiv:2006.09289, 2020.

Lee et al., “Regularized Autoencoders for Isometric Representation Learning”, ICLR, 2022.

Uscidda et al., “Disentangled Representation Learning with the Gromov-Monge Gap”, ICLR, 2025.

Shrestha et al., “Content-Style Learning from Unaligned Domains: Identifiability under Unknown Latent Dimensions”, ICLR, 2025.

Gui et al., “Multi-modal Contrastive Learning Adapts to Intrinsic Dimensions of Shared Latent Variables”, arXiv:2505.12473, 2025.

**Questions:**

1. Since SPLICE is a two-step procedure, which step is mainly responsible for the reported good performance in the experiments? What would happen if another method for disentangling shared-private latent variables is used for Step 1, together with L-ISOMAP in Step 2?
2. Did the authors try other manifold learning methods in place of L-ISOMAP for Step 2?
3. What are other potential applications for SPLICE, besides neuroscience?

---

> ### Author Response · Authors · 2025-11-24
> **Response to Reviewer NoRJ (1 of 2)**
>
> **Novelty:** We thank the reviewer for pointing out additional relevant work, which we have now included in the Related Work section of our discussion. We would like to note that the referenced papers, however, each only meet a subset of SPLICE’s strengths. We agree with the reviewer that there has been substantial previous work on multi-view private/shared disentangling, and separately, previous work on single-view geometry preservation. However, our work makes two novel contributions.
>
> Step 1 alone provides a significant advance in disentangling private/shared representations. The architecture we introduce, which adapts predictability minimization for multi-view shared-private disentangling, substantially outperforms leading methods in disentangling quality (Supp. Tables 1, 3, 4) and is robust to latent dimensionality mis-specification (Fig. 2d). This advance was not sufficiently highlighted in our initial submission – we have revised the text to more clearly highlight this, which we believe should be of interest within the disentangling literature per se.
>
> Combining multi-view private/shared disentangling with geometry estimation is non-trivial because in multi-view data, the shared and private geometries are entangled through nonlinear observation functions, so accurate estimation of each space’s geometry critically relies on high-quality disentangling. Improving disentangling quality was the primary motivation for developing the novel architecture that we present in Step 1 and SPLICE’s two-step architecture. Existing geometry-preservation methods infer a single latent manifold from the input data without performing shared-private separation, and existing shared–private models show poorer disentangling than our Step 1 and do not attempt to preserve manifold structure. **To our knowledge, SPLICE is the first method to jointly address both problems.** We have edited our discussion section to clarify these contributions.
>
> **Related work:** We are grateful to the reviewer for pointing us towards further related work. A new section in the discussion now cites these studies and describes how SPLICE is an advance over them. Although geometry-preserving loss terms have previously been used in autoencoder frameworks (Gropp et al and Lee et al), these methods are limited to single-view data and do not aim to disentangle groups of latent variables. Also in the single-view domain, Uscidda attempts to find a minimally distorting map from the observed data distribution to a disentangled isotropic Gaussian prior. However, as the authors note, the minimally distorting map may still necessarily introduce substantial geometric distortion when the distribution in observation space is not Gaussian, as is the case for most real data. SPLICE’s objective differs in that it applies to multi-view data, and aims to disentangle between shared and private latent spaces and preserve geometry within them.
>
> Regarding disentangling without prior knowledge of dimensionality, CLIP, which is analyzed in Gui et al., infers only shared latent variables. Shrestha et al. do learn statistically independent shared and private latent distributions without prior knowledge of dimensionality, but they are focused on data generation and use a GAN framework that does not include an encoder. Thus obtaining latents for individual samples, which is important for scientific analysis, requires gradient-based inversion of the generator w.r.t. its inputs (latents). This is quite slow for large numbers of samples; SPLICE explicitly learns the map from observation space to latent spaces via its encoders, and thus avoids this limitation. Furthermore, neither of these methods attempt to preserve the manifold geometry of their inferred latent spaces.
>
> **Quantitative results for Experiment 2:** We refer the reviewer to Supplementary Figures 9 and 10, which show the latent spaces of SPLICE and baselines for Experiment 2, along with R^2 values for linear decoders trained to predict ground truth latents from the inferred shared and private latents. SPLICE achieves near-perfect alignment with ground truth latent variables, vastly outperforming existing shared-only and shared-private methods. We have also added the R^2 number to the main text to make this more apparent.
>
> **Question 1:** Both steps are responsible for SPLICE’s performance. Our adversarial disentangling approach (step 1) alone disentangles more effectively than existing methods (Supp. Tables 1, 3, 4), and properly disentangled representations are necessary to produce valid submanifold projections that are input to the geometry-preserving fine-tuning in Step 2. Given that other disentangling methods empirically result in poorer disentanglement, using a different disentangling method for step 1 would result in contaminated and therefore less faithful representations of the geometry in step 2.

---

> > ### Author Response · Authors · 2025-11-24
> > **Response to Reviewer NoRJ (2 of 2)**
> >
> > **Question 2:** While we did not try other manifold learning techniques for Step 2, SPLICE requires only a matrix of pairwise distances between landmarks and each data point, so any geodesic estimation technique (robust versions of Isomap, LLE, MIND, etc) would fit seamlessly into SPLICE’s framework.
> >
> > **Question 3:** We appreciate this question, which allows us to highlight the broad applicability of our method. Interpreting the shared and private latent variables of multi-view data is a prevalent problem across main domains, including identifying the shared semantic overlap between text captions and images (Lee & Pavlovic, 2021), integrating information from single-cell transcriptomics and proteomics to characterize cell state (Argelaguet et al., 2021), and performing sensor fusion in robots (Fadadu et al., 2022). Because SPLICE does not incorporate any inductive biases that are highly specific to neural data, we believe that it is widely applicable to many domains.

---

> > > ### Comment · Reviewer_NoRJ · 2025-11-26
> > >
> > > I thank the authors for very detailed response.
> > >
> > > The positioning of this work is clearer now, and the authors should incorporate these clarifications into the manuscript. In addition, demonstrating applicability of SPLICE to domains beyond neuroscience might spark interests in other communities as well.
> > >
> > > I still have some reservation about the novelty at each step of SPLICE method.
> > >
> > > I will raise my score accordingly.

---

### Official Review · Reviewer_k3Xn · 2025-11-10

**Soundness:** 3
**Presentation:** 3
**Contribution:** 3
**Rating:** 8
**Confidence:** 3

**Summary:**

The manuscript presents SPLICE, a two-step neural network approach for unsupervised disentanglement in multi-view data, aiming to infer interpretable, non-linearly mixed shared and private latent representations while preserving their intrinsic submanifold geometry. The method first employs a crossed autoencoder and the predictability minimization objective for effective disentangling, and then refines the latents using a geometry-preserving loss derived from estimated geodesic distances along the submanifolds (calculated efficiently using landmarks). Evaluated on simulated CV data (rotated MNIST and SPRITES), simulated neural activity, and a real-world neurophysiological dataset, SPLICE yields interpretable representations, and exhibits robustness to mis-specified latent dimensionality compared to state-of-the-art disentangling and shared-only methods.

**Strengths:**

- Disentanglements include robustness to misspecification of latent dimensions as well as the loss regularization parameter $\lambda_{geo}$.
- Disentanglements show clearer separation compared to baselines and superior interpretability via geometry preservation.
- Rigorous experimental analysis spans well defined simulations covering rotated MNIST, SPRITES as well as simulated and real neural spiking datasets.

**Weaknesses:**

- Computational Efficiency:
  - How does the runtime of SPLICE compare to the other baseline approaches?
- Clarity:
  - The methodology in Step 2 - Geometry Identification and Preservation -  requires clarification.
  - Which method did you use to estimate the nearest neighbor graph?
- Hyperparameter Determination and Stability:
  - Clarify the rationale for taking multiple gradient steps for the measurement networks per autoencoder update, as mentioned in the manuscript. The hyperparameter for this is currently missing.
  - Algorithms 1 and 2 suggest frequent resetting of the measurement networks. Please discuss the convergence stability for practitioners.
  - State whether a fixed parameter set of $n_{msr}$ and $T_{restart}$ worked effectively across all datasets or if extensive tuning was required.

**Questions:**

see weaknesses

---

> ### Author Response · Authors · 2025-11-24
> **Response to Reviewer k3Xn**
>
> **Computational Efficiency:** SPLICE has a slightly longer runtime than pure disentangling methods; Step 1 is actually faster than competing methods for pure disentangling due to requiring less inner steps/less complex loss objectives than Lyu et al. 2021 and Lee & Pavlovic, 2021. The additional fine-tuning step (Step 2) required to achieve geometry preservation then makes the total time slightly more than pure disentangling methods. We believe that the additional compute time is small enough to be worth the gains in disentangling quality (crucial to validity of inferred latents) and interpretability (crucial to gaining insight into meaning/structure of inferred latents).
>
> **Clarity of Projection section:** We thank the reviewer for this comment – the two-step approach is a key design choice in our method. The shared and private geometries are entangled through nonlinear observation functions, so accurate estimation of each space’s geometry critically relies on high-quality disentangling. Given the properly disentangled representations from Step 1, we hold the private latents constant and vary the shared latents according to the training data. We then project the resulting latent combinations through the trained decoder. Because the projections were generated by holding private latents constants and varying only the shared latents, they isolate the effect of varying shared information in the data space – i.e. the shared submanifold. We can then use standard manifold learning techniques (nearest-neighbor graph construction -> geodesic estimation -> learn a geodesic-preserving embedding) to fine-tune the Step 1 networks to match the submanifold geometry. (An analogous process can be done for the private submanifold and the shared and private submanifolds for the other view). We have updated the methods sections to be more clear..
>
> **Nearest Neighbor Estimation:** We used the sklearn nearest neighbors implementation in Python, which uses the KDTree algorithm by default.
>
> **Measurment Network Hyperparameters:** We thank the reviewer for this question, as this design choice enables SPLICE’s superior disentangling performance. Taking multiple measurement network gradient steps per autoencoder update ensures that the measurement networks are well-trained, and thus able to detect any leakage that the encoders may introduce. We used the same values of $n_{msr}=5$ for all experiments, which we show to achieve very good disentangling. We did not experiment with different values, but a smaller $n_{msr}$ may also be sufficient. We have added a new paragraph to our discussion section to state which hyperparameters were constant across datasets and which were tuned.
>
> **On Cold Restarts:** We thank the reviewer for bringing this to our attention. While an earlier version of our method performed cold restarts to (in theory) prevent the measurement networks from settling into local minima and ignoring some leakage, we found that for the datasets in our paper, resetting did not improve disentangling quality in practice. We have therefore removed the resetting-related lines from Algorithms 1 and 2, but will retain measurement network resetting as an optional feature in our codebase.
>
> **Hyperparameter tuning:** We thank the reviewer for this comment. Although we originally mentioned hyperparameter and tuning details in the Supplementary Material, this information was not clear from the main text, and some details were missing from the supplement. We have added information to our discussion to clarify in the main text which parameters typically require tuning, and which ones were left static across datasets.

---

### Author Response · Authors · 2025-11-24
**Global Response to Reviewers**

We would like to thank all the reviewers for their helpful comments and questions. Our submission develops a new approach for multi-view data, both disentangling latent variables (with better quality than leading existing methods), and preserving intrinsic geometries of each of the private and shared latent representations. **To our knowledge, SPLICE is the first method to jointly address both problems.** All prior methods either 1) address only single view data (Gropp et al., 2020; Lee et al., 2021)  2) do not include geometry preservation (Lee & Pavlovic, 2021; Lyu et al., 2021; Shrestha et al., 2025), or 3) only implicitly disentangle, without explicit costs to enforce independence (Gondur et al., 2023; Sani et al., 2024). This includes studies that reviewers pointed us to, which we now cite and discuss in the manuscript text. SPLICE addresses all these concerns, and even outperforms on pure disentangling (i.e., without the geometry preservation) as compared to the leading methods.

**Novelty:** We would like to highlight two novel contributions of our work:

*Disentangling per se*. Step 1 of our algorithm alone (before geometry preservation) provides a significant advance in disentangling private/shared representations. The architecture we introduce, which adapts predictability minimization for multi-view shared-private disentangling, substantially outperforms leading methods in disentangling quality (Supp. Tables 1, 3, 4) and is robust to latent dimensionality mis-specification (Fig. 2d).

*Combining multi-view private/shared disentangling with geometry estimation.* Combining the two techniques is non-trivial because in multi-view data, the shared and private geometries are entangled through nonlinear observation functions, so accurate estimation of each space’s geometry critically relies on high-quality disentangling. Existing geometry-preservation methods infer a single latent manifold from the input data without performing shared-private separation, and existing shared–private models show poorer disentangling than our Step 1 and do not attempt to preserve manifold structure. Geometry estimation is especially important for blind scientific discovery, when we do not know the content of the shared and private latent spaces beforehand, and must rely on their geometric structure and effect on observations to deduce their content.

**Clarity of Methods:** Multiple reviewers pointed out that the Methods text about the projection step specifically was unclear. We have revised the methods text to make the motivation and process for the projections easier to understand.

We have addressed other comments/questions in the individual responses to each reviewer, and we have uploaded an updated PDF of our paper with new revisions in blue.

---

### Author Response · Authors · 2025-11-30
**Summary of Discussion Period for New AC**

Dear (new) Area Chair,

Thank you for the time and effort in reviewing and working with a whole new stack of papers given the unfortunate Openreview leak. To hopefully alleviate the burden on your time we would like to summarize the interactions that we have had with the reviewers to date. The initial review was positive, with one holdout (88662; Reviewer NoRJ giving the 2). The reviewers had a number of helpful suggestions and points of clarification. In response we had responded and/or edited our manuscript to:
1. Better emphasize the lack of prior work in algorithms that both disentangle private and shared information while also preserving geometry (Revs NoRJ, R5mr)
1. Add emphasis that even without the geometry preservation, the first step in our algorithm practically disentangles better than current state-of-the-art methods, indicating that even this step alone is a valuable advance (Rev NoRJ)
1. Add an additional experiment using a different encoding decoding architecture (convolutional layers instead of fully connected layers) to show that the results do not reflect architecture-specific benefits (Rev UzvA)
1. Clarify the minimal additional runtime of the algorithm that is needed for the retraining in the geometry preservation, and that this step is crucial to interpretability and worth the mild additional runtime (Rev k3Xn)
1. Clarify hyperparameter tuning and cold restarts (Revs k3Xn, P8Fy)

Following our response we had received positive (thankfully early) feedback from the reviewers:
- **Reviewer UzvA** thanked us for clarifications and **raised their score from a 6 to an 8**
- **Reviewer P8Fy** thanked us for clarifications and stated that they would **maintain their positive score of 8**
- **Reviewer NoRJ** thanked us for the clarifications, and stated that they better appreciate the combined disentangling and geometry preservation. They stated in the response that they **would increase the score to >2**, however the final score never came through and they did not state what they would raise it to.

The revisions to our manuscript are colored blue in the PDF. We hope that this summary appropriately captures the history of the conversation up to the time that the discussion period was cut short. We are happy to clarify any point and once again appreciate the extra work that the current situation is requiring of the ACs.

Warmest regards,

The authors

---

### Meta-Review · Area_Chair_zgJf · 2025-12-19

**Summary:**

This paper proposes a multi-view approach for disentanglement. Reviewer NoRJ cited a lack of discussion and comparison with prior methods, which, unfortunately, is far more severe than the reviewer pointed out. In fact, the paper ignores a large body of works in the disentanglement and causal represetation learning communities looking at multi-view data. For example, in chronological order:

[1] Kulkarni et al., "Deep Convolutional Inverse Graphics Network", NeurIPS 2015
[2] Withney et al., "UNDERSTANDING VISUAL CONCEPTS WITH CONTINUATION LEARNING", ICLR workshops 2016
[3] Feng et al., "Dual Swap Disentangling", NeurIPS 2018
[4] Kim et al., "Disentangling by Factorising", ICML 2018
[5] Bouchacourt et al., "Multi-level variational autoencoder: Learning disentangled representations from grouped observations", AAAI 2018
[6] Gresele et al., "The incomplete rosetta stone problem: Identifiability results for multi-view nonlinear ica.", UAI 2020
[7] Locatello et al., "Weakly supervised disentanglement without compromises", ICML 2020
[8] Fumero et al., "Learning disentangled representations via product manifold projection", ICML 2021
[9] Bremher et al., "Weakly supervised causal representation learning", NeurIPS 2022
[10] Ahuja et al., "Weakly supervised representation learning with sparse perturbations", NeurIPS 2022
[11] Bereket et al., "Modelling Cellular Perturbations with the Sparse Additive Mechanism Shift Variational Autoencoder", NeurIPS 2023

Many papers proposed a very similar problem setup and solution. [1] used multiple images where certain attributes stay the same and the representation is then averaged. [2] does a similar thing but imagining the images come from time series (two consecutive frames) and the image is decoded swapping latents between the two with a learned gating mechanism (extremely similar to this work). [3] also does the swapping between two images, but in an autoencoder, not a VAE (also, extremely similar to this work). [5] used grouped observations and modeled the sharing of components in a Bayesian framework using VAEs. [6] provides identifiability result for exactly this multi-view setup in the context of non-linear ICA, and so does [7] for disentanglement with a very similar algorithm than the one proposed in this submission, but averaging the shared components instead of swappping. [8] specifically look at submanifolds. Predictability minimization for disentanglement was already used by [4]. In fact, the concern raised by the reviewer UzvA that the method is heuristic is invalid; this problem has extensive identifiability results [6,7], which are unfortunately already known.

The paper should compare its method against the literature, as it is currently severely misplaced. For example, [7] is extremely close to the setting studied and should be compared against (should be about a one line of code change compared to the proposed method). Also, a number of data sets have been used for this task. If the authors want to propose this method as a new way to do disentanglement, they should also compare it on the datasets that already exist. The new dataset from neurophysiological experiments is really nice and would be a good addition to the field.

Frankly, I would recommend the authors to re-position the paper around the data set and the use case, as both the methodology and the setting of using grouped/paired observations have a very long tradition in disentanglement. As of now, I need to unfortunately recommend rejection.

**Reviewer Concerns:**

The main concern that the reviewers raised that I am using as an argument for rejection is the positioning with respect to the literature (NoRJ, R5mr). The review from k3Xn was brief and shallow. These were not addressed by the authors convincingly, as they essentially have "brushed them away". They should have taken the chance to do a deep dive in the literature and realize they had missed large parts of the literature.

**Reviewer Scores:**

The reviewer's score is largely irrelevant in this discussion. I believe that if I had the chance to engage with the reviewers in the discussion they would have lowered their scores.

---

### Decision · Program_Chairs · 2026-01-26

Reject